# Uncover Governing Law of Pathology Propagation Mechanism Through A Mean-Field Game

**Tingting Dan**      **Zhihao Fan**      **Guorong Wu**[*]
Departments of Psychiatry and Computer Science
University of North Carolina at Chapel Hill
Chapel Hill, NC 27599
{Tingting_Dan,grwu}@med.unc.edu;zhihaoffan@gmail.com

## Abstract

Alzheimer's disease (AD) is marked by cognitive decline along with the widespread of tau aggregates across the brain cortex. Due to the challenges of imaging pathology spreading flows *in vivo*, however, quantitative analysis on the cortical pathways of tau propagation and its interaction with the cascade of amyloid-beta (A$\beta$) plaques lags behind the experimental insights of underlying pathophysiological mechanisms. To address this challenge, we present a physics-informed neural network, empowered by mean-field theory, to uncover the biologically meaningful spreading pathways of tau aggregates between two longitudinal snapshots. Following the notion of 'prion-like' mechanism in AD, we first formulate the dynamics of tau propagation as a mean-field game (MFG), where the spread of tau aggregate at each location (aka. agent) depends on the collective behavior of the surrounding agents as well as the potential field formed by amyloid burden. Given the governing equation of propagation dynamics, MFG reaches an equilibrium that allows us to model the evolution of tau aggregates as an optimal transport with the lowest cost in *Wasserstein* space. By leveraging the variational primal-dual structure in MFG, we propose a *Wasserstein*-1 Lagrangian generative adversarial network (GAN), in which a Lipschitz critic seeks the appropriate transport cost at the population level and a generator parameterizes the flow fields of optimal transport across individuals. Additionally, we incorporate a symbolic regression module to derive an explicit formulation capturing the A$\beta$-tau crosstalk. Experimental results on public neuroimaging datasets demonstrate that our explainable deep model not only yields precise and reliable predictions of future tau progression for unseen new subjects but also provides a new window to uncover new understanding of pathology propagation in AD through learning-based approaches.

## 1   Introduction

Alzheimer's disease (AD) is marked by a progressive decline in cognition accompanied by widespread accumulation of tau aggregates across the cortex. Mounting evidence suggests that tau spreads in a 'prion-like' fashion: once a small number of molecules misfold, they act as seeds that affect neighboring neurons, propagating through neural circuits much like a contagion. In parallel, extracellular amyloid, beta (A$\beta$) plaques, often accumulating years before symptom onset, are known to prime neural tissue by promoting tau hyperphosphorylation and enhancing trans-synaptic spread [23; 5]. Together, these two hallmarks of AD form a toxic synergy that accelerates protein aggregation, neuronal damage, and memory loss. Yet, the precise cortical pathways along which tau propagates,

---

[*]Corresponding author.

39th Conference on Neural Information Processing Systems (NeurIPS 2025).

and how A$\beta$ modulates or accelerates those flows, remain open questions in neurodegeneration research [3; 24; 30; 18; 17; 36].

Early theoretical work inspired by epidemiology and chemical kinetics formulates this process as a reaction–diffusion system, in which tau both drifts along concentration gradients and undergoes nonlinear local amplification. For instance, Iturria-Medina et al. [23] demonstrated that a partial differential equation (PDE) with an explicit reaction term accurately reproduces the spatiotemporal patterns seen in longitudinal tau-PET scans. Crucially, extracellular amyloid-$\beta$ plaques, often present years before clinical onset, have been shown to "prime" neural circuits, enhancing tau phosphorylation and accelerating its trans-synaptic spread [5]. This synergistic interplay drives a vicious feed-forward loop of protein aggregation and neuronal damage, underscoring the necessity of models that capture the interaction between amyloid plaques and tau aggregates.

In general, there are five popular approaches to capture these intertwined dynamics. (1) *Reaction–Diffusion Models (RDM)*. Building on prion-like hypotheses, continuous reaction–diffusion equations capture both the drift of tau along spatial gradients and its local nonlinear accumulation [23]. Such models assume homogeneous kinetics and ignore complex network geometry. (2) *Connectome-Based Network Diffusion*. By projecting tau as a density on the structural connectome, Raj et al. [35] used a linear graph-diffusion operator to simulate tau transport along white-matter tracts. Their network diffusion model (NMD) accurately predicted regional atrophy patterns across cohorts but treats the brain as a passive conduit without explicit reaction kinetics. (3) *Graph Reaction–Diffusion*. Extending network diffusion, Vogel et al. [39] introduced nonlinear reaction terms on graph Laplacians to jointly model diffusion and local tau–amyloid interactions on anatomical networks. While more expressive, these methods still rely on hand-tuned reaction laws and lack end-to-end learning of reaction kinetics. (4) *Data-Driven Deep Learning*. Recent work harnesses convolutional neural networks (CNNs) to learn tau progression directly from imaging data. Lee et al. [29] trained CNNs on positron emission tomography (PET) sequences to forecast future tau maps but found these black-box models often overfit and offer limited mechanistic insight. (5) *Graph Neural Networks (GNN)*. Recent works [2; 14] leveraged GNNs to capture both network topology and nonlinear interactions, showing improved regional predictions. Due to the 'black-box' nature, however, it is challenging to generate interpretable governing laws through GNN only. From a system-level perspective, current approaches simply assume the evolution of tau propagation following a pre-defined physics model, without actively identifying or optimizing the most suitable governing principle for the underlying dynamics.

Notably, nearly all existing tau-propagation models [40; 21; 16; 10] operate at the level of coarse anatomical regions or volumetric parcels, effectively "down-sampling" the cortex into a handful of nodes and treating each as spatially homogeneous. Although region-based graphs offer computational efficiency from a modeling perspective, such oversimplification of the cortical sheet's fine-grained geometry, such as folds, sulci, and gyri, limits their ability to accurately capture how misfolded proteins diffuse and interact at the voxel resolution. In contrast, surface-based PET studies [22; 15] have shown the potential to address this limitation. For example, Xia et al. [42] projected [18F]-AV-1451 uptake onto individual FreeSurfer surfaces and demonstrated that tau spreads in waves across temporal and parietal gyri, following the cortex's folds rather than simple volumetric adjacency. Cho et al. [9] extended this to a two-year longitudinal analysis with [18F]-flortaucipir, revealing concentrated tau accumulation in medial, basal, and lateral temporal regions and clear propagation trajectories along the surface. In light of this, our proposed model is built directly on the cortical surface mesh, with >160,000 vertices that faithfully trace the brain's highly convoluted topology.

Taken together, we propose a physics-informed deep learning framework that unites biophysical modeling and data-driven discovery to reconstruct tau propagation dynamics from longitudinal tau-PET scans. *First*, we formulate tau spread as a mean-field game (MFG), where each cortical location (agent) evolves under the influence of a local tau–amyloid interaction field and the collective behavior of neighboring regions. This variational formulation naturally induces an optimal transport process in *Wasserstein* space, capturing both reaction and diffusion within a theoretically grounded structure. *Second*, the forward-backward structure in MFG naturally leads to a saddle point formulation within a min-max optimization framework [1]. As shown in Fig. 1, we design a *Wasserstein*-1 Lagrangian generative adversarial network (GAN), where a generator learns subject-specific tau velocity fields and a Lipschitz critic estimates population-level transport costs. *Third*, we incorporate symbolic regression to learn an explicit, interpretable tau–A$\beta$ reaction law directly from data. Unlike previous region-based or volumetric models, our model operates directly on the cortical surface mesh, leveraging over 100k vertices to capture fine-grained geodesic flows along sulci and gyri. This

anatomical fidelity enables our model to uncover biologically plausible propagation pathways and mechanistic insight into tau–amyloid interplay. Our experiments demonstrate that our MFG-based deep model not only delivers precise and reliable predictions of future tau accumulation but also reveals interpretable dynamics aligned with neuropathological staging and recent imaging studies.

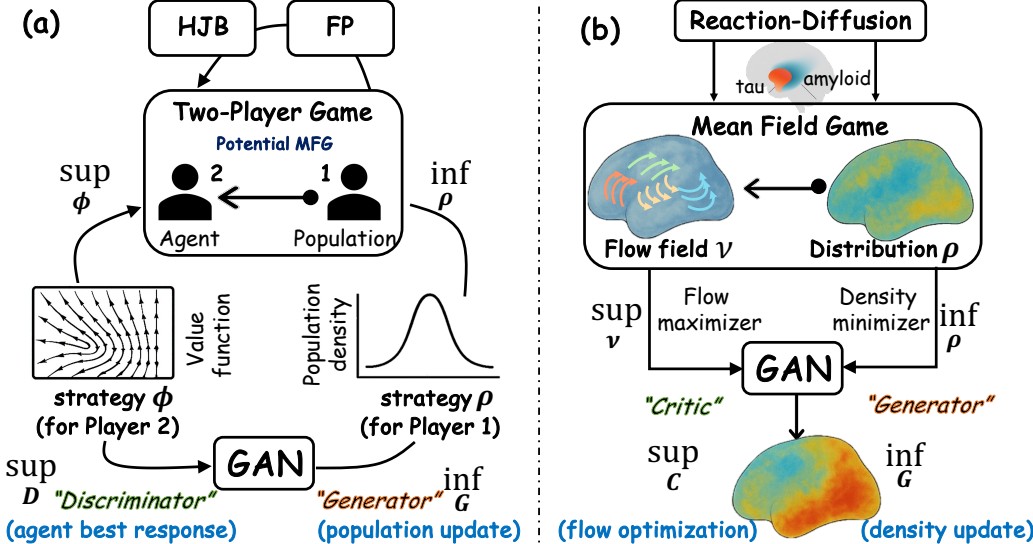

Figure 1: Schematic sketch for the methodological connection between RDM, MFG, GAN in our model. (a) A potential mean field game is obtained by coupling the Hamilton–Jacobi–Bellman (HJB) and Fokker–Planck (FP) equations into a two-player saddle-point formulation: Player 1 (population density $\rho$) minimizes via $\inf_\rho$, while Player 2 (value function $\phi$) maximizes via $\sup_\phi$. This variational game underpins zero-sum optimal transport dynamics. (b) By casting cortical tau spreading as a reaction–diffusion process on the brain cortex, one recovers an equivalent deterministic MFG between the flow field $\nu$ and tau density distribution $\rho$. Using the Kantorovich–Rubinstein dual of the Earth–Mover (*Wasserstein*-1, $W_1$) distance, we formulate a $W_1$–Lagrangian GAN: the Critic (*Flow Maximizer*, $\sup_v$) learns the transport cost, and the Generator (*Density Minimizer*, $\inf_\rho$) predicts the next-time tau distribution $\hat\rho_{t+1}$. Alternating these updates unifies PDE-based flow optimization and data-driven density forecasting.

## 2 Methods

**Data Description.** We can organize brain cortex data as a graph $\mathcal{G} = (X, D)$, where $X = \{x_i \mid i = 1, \dots, N\}$ represents a set of $N$ predefined cortical locations (e.g., surface parcels or mesh vertices), and $D = [d_{ij}]_{i,j=1}^N$ contains the Euclidean distances between all vertex pairs, with $d_{ij} = \|x_i - x_j\|$. At each cortical site $x_i$, we obtain two time-varying scalar standardized uptake value ratios (SUVR): $u(t) = [u_1(t), \dots, u_N(t)]^\top$, $v(t) = [v_1(t), \dots, v_N(t)]^\top$ and $u(t+1) = [u_1(t+1), \dots, u_N(t+1)]^\top$, corresponding to longitudinal measurements of tau and amyloid concentrations, respectively.

### 2.1 How Brain Proteins Travel: Insights from Diffusion and Game Theory

**Reaction Diffusion Model.** Tau protein propagation in Alzheimer's disease often resembles the way a drop of ink spreads in water, but constrained by the intricate folding of the cortex. To capture this, we postulate a *reaction–diffusion* model (RDM) (partial differential equation, PDE) on our cortical graph:

$$\frac{\partial u(t)}{\partial t} = S\big(u(t)\big) + R\big(u(t),\, v(t)\big), \tag{1}$$

▶ $S(u)$ is the *diffusion operator*, modeling pure tau spread along the network topology (brain cortex);

▶ $R(u, v)$ is the *reaction operator*, capturing the interaction between tau and amyloid.

Physically, $S(u)$ governs how tau "leaks" between neighboring regions (much like heat conduction), while $R(u, v)$ governs how amyloid burden might accelerate or inhibit tau accumulation.

In a typical machine-learning implementation, the diffusion term $S(u)$ can be instantiated by a graph neural network (GNN) that learns to approximate the action of the Laplacian $-\nabla \cdot (\nabla u)$. However,

using a standard multilayer perceptron (MLP) for $R(u, v)$ often yields a black-box model with limited interpretability. To address this, we replace the MLP with a *symbolic regression* module [34], which discovers an explicit algebraic formula for $R$. The outcome is a hybrid framework that: (1) Leverages GNNs for accurate, geometry-aware diffusion $S(u)$; and (2) Uses symbolic regression to yield a transparent, human–readable reaction law $R(u, v)$.

**From Reaction–Diffusion to Mean Field Games.** Interestingly, the same reaction–diffusion PDE (Eq. (1)) can be obtained as the optimality condition of a *potential* Mean Field Game (MFG) [31]. In that viewpoint, each infinitesimal "particle" of tau chooses a trajectory to minimize transport cost (diffusion) while experiencing local rewards or penalties from amyloid (reaction). Formally, this formulation boils down to a saddle–point problem:

$$\inf_{\rho(\cdot, 0) = u(0)} \sup_{\phi} \left\{ \int_0^T \int_\Omega \left[ \partial_t \phi + H(\nabla \phi) \right] \rho \, dx \, dt \; - \; \int_0^T \int_\Omega F\big(\rho(x, t), v(x, t)\big) \, dx \, dt \right\}, \quad (2)$$

where we choose $H(p) = \frac{1}{2} \|p\|^2, \qquad F(\rho, v) = - \int_0^\rho R(s, v) \, ds$, so that $\partial_\rho F(\rho, v) = -R(\rho, v)$. The corresponding Euler–Lagrange conditions are

$$\begin{cases} \text{(HJB):} & -\partial_t \phi + H(\nabla \phi) = -R(\rho, v), \\ \text{(FP):} & \partial_t \rho - \nabla \cdot \big(\rho \, \nabla_p H(\nabla \phi)\big) = 0. \end{cases} \quad (3)$$

Since $\nabla_p H(p) = p$, the Fokker–Planck equation becomes $\partial_t \rho - \nabla \cdot (\rho \, \nabla \phi) = 0$. Now assume a *uniform density* $\rho(x, t) \equiv 1$ and identify the value function $\phi$ with the tau concentration $u$. Then

$$\partial_t u = -\nabla \cdot (\nabla u) + R(u, v), \quad (4)$$

which exactly reproduces the reaction–diffusion PDE (Eq. (1)). Thus, tau spreading on the cortex can be viewed both as a network-constrained reaction–diffusion process and as the Nash equilibrium of a deterministic potential MFG. Because such MFGs admit the Kantorovich–Rubinstein dual [38], this duality naturally connects our reaction–diffusion formulation to the GAN-driven optimal transport framework.

## 2.2 GAN-Driven Flow Field Evolution Using *Wasserstein*-1 Metrics and Lagrangian Principles

**Problem Formulation.** Let $\mathcal{X} = \{x_i\}_{i=1}^N \subset \mathbb{R}^d$ denote cortical coordinate domain (e.g., cortical mesh). At each time $t$, we observe the tau concentration vector $u(t)$, which defines an empirical distribution $\rho_t \in \text{Prob}(\mathcal{X})$. Our goal is to learn both: (1) a *flow field* $\nu(x, t)$ on the mesh that drives tau transport, (2) and the predicted *tau density* $\hat{u}(t + 1)$, so that $\hat{u}(t + 1) \approx \rho_{t+1}$, the true next-step distribution. The classical optimal mass transport (OMT) formulation, under the squared-$\ell_2$ cost (*Wasserstein*-2), is given by

$$\inf_{\substack{\rho(\cdot, 0) = \rho_t \\ \rho(\cdot, 1) = \rho_{t+1}}} \int_0^1 \int_\mathcal{X} \frac{\|q(x, s)\|^2}{\rho(x, s)} \, dx \, ds, \quad \text{s.t. } \partial_s \rho + \nabla \cdot q = 0, \quad (5)$$

where $q(x, s) = \rho(x, s) \, \nu(x, s)$ is the flux field. This yields $W_2(\rho_t, \rho_{t+1})$ but requires discretizing the "pseudo-time" $s \in [0, 1]$. By contrast, the Earth–Mover (*Wasserstein*-1) distance

$$W_1(\rho_t, \rho_{t+1}) = \inf_{\gamma \in \Pi(\rho_t, \rho_{t+1})} \mathbb{E}_{(x, y) \sim \gamma} \big\| x - y \big\| \quad (6)$$

admits the Kantorovich–Rubinstein dual [38]:

$$W_1(\rho_t, \rho_{t+1}) = \sup_{\|C\|_L \leq 1} \left\{ \mathbb{E}_{x \sim \rho_t}[C(x)] \; - \; \mathbb{E}_{y \sim \rho_{t+1}}[C(y)] \right\}. \quad (7)$$

This dual form can (1) avoid discretizing an extra "time" variable, (2) provide a Lipschitz-constrained critic $C$ that yields smoother, more stable gradients, (3) remain well-posed even when $\rho_t$ and $\rho_{t+1}$ have disjoint support.

**Wasserstein$_1$–Lagrangian GAN for Flow Evolution.** Building on the dual formulation (Eq. 7), we cast tau spreading as a two-player adversarial game in which one network infers the flow field that transports the current tau distribution $\rho_t$ into the next distribution $\rho_{t+1}$.

▶ *Generator $G_\theta$ (Density Predictor).* Imagine "pulses" of tau flowing across the brain's surface under the combined influence of diffusion and local biochemical reactions (as shown in Fig. 2, a). To do so, $G_\theta$ is a *reaction–diffusion engine* (see Sec. 2.3 for details), which, given the current tau/amyloid state $\big(u(t), v(t)\big)$, computes the flow field $\nu = G_\theta\big(u(t), v(t)\big)$ and advances tau by one Lagrangian step of size $\Delta t$. The result $\hat{u}(t + \Delta t)$ induces the "push-forward" measure on the mesh $\hat{\rho}_{t+1} = \big(x + \nu\,\Delta t,\ \hat{u}(t + \Delta t)\big)_{\#}\,\rho_t$.

▶ *Critic $C_\varphi$ (Flow Maximizer).* Let $C_\varphi : \mathcal{X} \to \mathbb{R}$ be a neural network with parameters $\varphi$, constrained so that its Lipschitz constant satisfies $\|C_\varphi\|_L \leq 1$. The critic's objective is to *maximize* the estimated Earth–Mover gap $\mathcal{L}_C(\varphi) = \mathbb{E}_{x \sim \hat{\rho}_{t+1}}\big[C_\varphi(x)\big]\ -\ \mathbb{E}_{y \sim \rho_{t+1}}\big[C_\varphi(y)\big]$. By pushing $C_\varphi$ to increase this difference under the 1-Lipschitz constraint, the critic approximates the *Wasserstein–1 distance* $W_1(\rho_t, \rho_{t+1})$ between the generated and true tau distributions.

The generator then *minimizes* this critic score on its own prediction: $\mathcal{L}_G(\theta) = \mathbb{E}_{x \sim \hat{\rho}_{t+1}}\big[C_\varphi(x)\big]$.

Together, they play the saddle-point game

$$\inf_\theta \sup_{\|\varphi\|_L \leq 1} \Big\{ \mathbb{E}_{x \sim \hat{\rho}_{t+1}}\big[\,C_\varphi(x)\,\big]\ -\ \mathbb{E}_{y \sim \rho_{t+1}}\big[\,C_\varphi(y)\,\big] \Big\}. \tag{8}$$

**Connections to MFGs**. To see why our *Wasserstein$_1$–Lagrangian GAN* critic solves a deterministic MFG, we interpolate between $\rho_t$ and $\rho_{t+1}$ over a "pseudo-time" $s \in [0, 1]$. Let $\rho(x, s), \quad s \in [0, 1]$, satisfy the continuity equation $\partial_s \rho + \nabla \cdot (\rho\,\nu) = 0, \qquad \rho(\cdot, 0) = \rho_t, \quad \rho(\cdot, 1) = \rho_{t+1}$, where $\nu(x, s)$ is the velocity (flow) field. A *potential* MFG formulation is then the saddle-point problem

$$\inf_{\rho(\cdot, s)} \sup_\phi \int_0^1 \int_{\mathcal{X}} \big[\partial_s \phi + H(\nabla \phi)\big]\, \rho\, dx\, ds\ -\ \mathcal{T}\big(\rho(\cdot, 1)\big), \tag{9}$$

with terminal constraint enforced by $\mathcal{T}(\rho(\cdot, 1))$ so that $\rho(\cdot, 1) = \rho_{t+1}$. Here: $\phi(x, s)$ plays the role of the *critic* or value-function. The Hamiltonian is the indicator $H(p) = \begin{cases} 0, & \|p\| \leq 1, \\ +\infty, & \|p\| > 1, \end{cases}$ which corresponds to the Lagrangian $\Lambda(\nu) = \|\nu\|$. The optimality (Euler–Lagrange) conditions are

$$\begin{cases} \text{(HJB)}: & -\partial_s \phi + H(\nabla \phi) = 0, \\ \text{(FP)}: & \partial_s \rho + \nabla \cdot (\rho\,\nabla \phi) = 0. \end{cases} \tag{10}$$

Since $H(p) = 0$ whenever $\|p\| \leq 1$, the HJB equation implies $\partial_s \phi = 0$ and hence $\phi(x, s) \equiv \phi(x)$. Substituting back and using only the terminal constraint $\rho(\cdot, 1) = \rho_{t+1}$ reduces the saddle point to

$$\sup_{\|\nabla \phi\| \leq 1} \Big\{ \underbrace{\int_{\mathcal{X}} \phi(x)\, \rho_t(x)\, dx}_{\mathbb{E}_{x \sim \rho_t}[\phi(x)]}\ -\ \underbrace{\int_{\mathcal{X}} \phi(x)\, \rho_{t+1}(x)\, dx}_{\mathbb{E}_{x \sim \rho_{t+1}}[\phi(x)]} \Big\}, \tag{11}$$

which is exactly the Kantorovich–Rubinstein dual for $W_1(\rho_t, \rho_{t+1})$. In our GAN, the critic $C_\varphi$ approximates this optimal $\phi$, and the generator $G_\theta$ seeks the flow field for which the push-forward $\hat{\rho}_{t+1}$ minimizes this same dual objective. Thus, the adversarial *Wasserstein$_1$–Lagrangian-GAN* training directly implements the equilibrium of a potential MFG, with the critic maximizing the Earth–Mover gap and the generator minimizing it. For clarity, Fig. 1 illustrates how the RDM, its equivalent potential MFG formulation, and the resulting *Wasserstein–Lagrangian GAN* are formally connected.

## 2.3  *MFG4AD*: A Physics-informed GAN for Modeling Tau Propagation in AD

**Network Architecture**. Building on the above link between reaction–diffusion, mean field games, and *Wasserstein–Lagrangian GANs*. The framework integrates reaction-diffusion modeling, symbolic regression, and GAN into a unified architecture, coined *MFG4AD*. We now drill into the *Generator* network architecture that powers *MFG4AD*. At the heart of *MFG4AD*, our generator $G_\theta$ is a *reaction–diffusion engine* tailored to the cortical mesh $\mathcal{G} = (X, D)$:

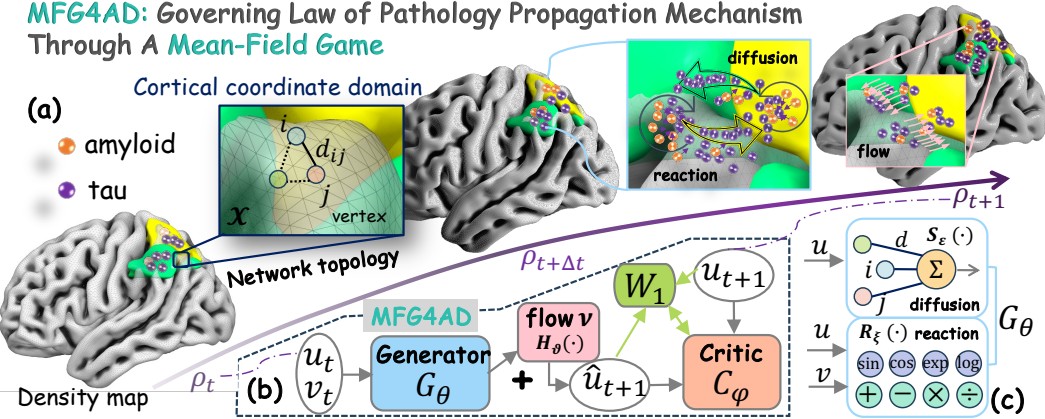

Figure 2: *MFG4AD*: A physics-informed deep learning framework for modeling tau propagation and amyloid-tau interactions in Alzheimer's disease. (a) We conceptualize the cortical surface as a graph, where vertices represent cortical locations and edges encode distances ($d_{ij}$). The tau concentration defines the initial and terminal density distributions, $\rho_t$ and $\rho_{t+1}$, respectively, while amyloid acts as an external modulator influencing the evolution of tau through a reaction term. (b) We propose a *MFG4AD*, consisting of a generator $G_\theta$ that predicts the next-time density ($\hat{u}_{t+1}$), and a critic $C_\varphi$ that evaluates the *Wasserstein*-1 distance between the predicted ($\hat{\rho}_{t+1}$) and true distributions ($\rho_{t+1}$). The optimal transport flow field $\nu$ guides the evolution of tau density. (c) The generator $G_\theta$ combines a GNN to model network-constrained diffusion (tau spreading along cortical pathways) and a symbolic regression module for explicit, interpretable tau–amyloid reaction dynamics.

¶ (1) Graph-based diffusion. We first leverage a graph neural network $S_\varepsilon$ (Fig. 2 (c), top) to approximate the Laplacian operator $-\nabla \cdot (\nabla u)$. Concretely, each vertex $x_i$ pools tau values $u(t)$ from its neighbors weighted by the geometry-$D$ and computes a discrete diffusion flux $P_i = S_\varepsilon(u(t), D)$, which captures how tau "leaks" along cortical folds.

¶ (2) Symbolic reaction. Next, we account for tau–amyloid crosstalk, which captures how amyloid catalyzes or inhibits tau. A symbolic regression module ($R_\xi$) (Fig. 2 (c), bottom) ingests the tau–amyloid pair $(u_i(t), v_i(t))$ at each vertex and outputs an explicit reaction rate $Q_i = R_\xi\big(u_i(t), v_i(t)\big)$.

¶ (3) Infer the flow field. We bundle each vertex's current tau level and its two physics-driven quantities into a state descriptor $F_i = \big[u_i(t), P_i, Q_i\big]$. A lightweight neural subnetwork $H_\vartheta$ then turns these descriptors into a movement vector $\nu_i \in \mathbb{R}^d$: $\nu_i = \big[H_\vartheta(F)\big]_i$. Finally, we let each bit of tau ride this flow field via one Lagrangian step of size $\Delta t$:

$$\hat{u}_i(t + \Delta t) = u_i(t) + \Delta t \Big[-\mu_1 \nabla \cdot \big(u(t)\,\nu(t)\big)\big|_{x_i} + \mu_2\,P_i + \mu_3\,Q_i\Big], \tag{12}$$

where $\mu_1, \mu_2, \mu_3$ are the learnable scalars that let the model automatically tune the relative strengths of diffusion, reaction, and source contributions to best match longitudinal tau data. By doing so, $G_\theta$ is not a black box but a *reaction–diffusion engine* that explicitly computes diffusion, reaction, and advection to predict the next tau map $\hat{u}(t + \Delta t)$. Then a "push-forward" derives empirical measure $\hat{\rho}_{t+1}$ on the mesh, which the *Critic* evaluates against the true distribution $\rho_{t+1}$ under the *Wasserstein*-1 metric. Training alternates between optimizing these two networks (Fig. 2, b), allowing the generator to learn biologically meaningful, accurate predictions of tau propagation, while the critic ensures stable convergence by evaluating the quality of the generated densities.

**Training Phase.** We summarize our training procedure in Algorithm 1 (as shown in Appendix). Each generator update is preceded by $n_C = 5$ critic updates, with learning rates set to $\eta_C = 1 \times 10^{-5}$ for the critic and $\eta_G = 1 \times 10^{-4}$ for the generator. To enforce the 1-Lipschitz constraint on the critic $C_\varphi$, we apply spectral normalization to every layer [32]. The generator's loss combines the adversarial term with an $\ell_1 = |\hat{u}_i(t+1) - u_i(t+1)|_1$ reconstruction penalty weighted by $\lambda = 10$, ensuring both realistic and accurate predictions.

## 3 Experiments

**Data Preprocessing**. *Tau/Aβ SUVR Generation*. We process each subject's T1-weighted (T1W) MRI with FreeSurfer to reconstruct the cortical surfaces (white, pial, and mid-thickness) and to define

cerebellar gray matter as our reference region. Next, we rigidly register the motion-corrected tau-PET and amyloid-PET volumes to the T1W image, resample them into MRI space, and compute voxel-wise SUVR by dividing each voxel's uptake by the mean signal in the cerebellar reference. These SUVR volumes are then projected onto the subject's pial surface ($\sim$100k vertices) via trilinear interpolation and lightly smoothed along the mesh. To facilitate group analysis, each surface SUVR map is warped into the MNI template (fsaverage in MNI space), resampled onto the same 163,842-vertex mesh, and z-score normalized across cortical vertices. The resulting high-resolution, surface-based SUVR profiles at times $t$ and $t + 1$ for tau, and at time $t$ for amyloid, constitute the inputs $u(t), v(t), u(t+1)$ for our *MFG4AD*. *Network Topology Construction*. To capture the anatomically faithful geometric relationships among cortical vertices, we construct a sparse undirected graph directly from the native pial-surface mesh generated by FreeSurfer [13]. Each vertex is treated as a graph node, and edges are defined according to the triangular tessellation of the cortical surface: two nodes are connected if they share an edge in the mesh. This results in a biologically grounded graph structure with an average node degree of approximately 6, preserving submillimeter-scale geometry while adhering to the true cortical topology. Rather than computing a full pairwise distance matrix, we leverage the mesh's intrinsic sparsity, storing only the anatomical edges defined by surface adjacency. This allows diffusion operations to scale in $\mathcal{O}(kN)$ time, where $k$ is the average vertex degree. To simulate tau propagation, we implement the reaction–diffusion step via vectorized sparse-tensor operations over this mesh-defined graph, enabling a full forward Euler step across >100k cortical vertices in milliseconds. The full mesh-based construction process is described in Appendix A.1.

**Experimental Setup**. We evaluate the performance of *MFG4AD* using two longitudinal tau PET datasets: the Alzheimer's Disease Neuroimaging Initiative (ADNI) [25] and the Open Access Series of Imaging Studies (OASIS) [28]. The ADNI dataset includes 134 participants with both tau and amyloid PET scans, each with 2–6 longitudinal visits, resulting in a total of 631 scan pairs. Subjects in ADNI are categorized into five diagnostic groups: cognitively normal (CN), subjective memory complaint (SMC), early mild cognitive impairment (EMCI), late mild cognitive impairment (LMCI), and AD. The OASIS dataset comprises 77 participants, each with two longitudinal PET scans, and includes two diagnostic groups: CN and AD. Together, these two datasets provide a diverse and representative sample across the Alzheimer's disease spectrum, enabling comprehensive evaluation of our predictive framework across multiple disease stages. Comparative methods span five categories: (1) Connectome-Based Network Diffusion Model, NDM [35]. (2) *Graph-Based Methods:* vanilla GCN [41] and the advanced GCNII [7]. (3) *Deep Learning Models:* deep neural networks (DNN) composed of MLPs, deep symbolic model (DSM) [26] and vanilla GAN [19]. (4) *PDE-Based Methods:* graph neural diffusion (GRAND) [6], Neuro-ODE [8] and graph neural reaction-diffusion networks (GREAD) [11]. (5) *Traditional Regression Model:* Ridge regression (a regularized linear regression model). For all experiments, we conduct 5-fold cross-validation. The evaluation metrics for testing results include: mean absolute error (MAE) and root mean squared error (RMSE), between the predicted tau burden and the observed tau SUVR from follow-up PET scans. All models are trained for 1,000 epochs with Adam [27] optimizer.

### 3.1 Model Behavior and Ablation Study

**Prediction Performance of Future Tau Accumulation.** To evaluate the predictive performance of different computational approaches, we used baseline tau concentration and combined tau + amyloid as two types of input, with follow-up tau SUVR measurements serving as the ground truth. Prediction errors for each method are summarized in Table 1, with results from ADNI and OASIS shown on the left and right, respectively. Experimental findings demonstrate that our proposed method consistently outperforms all competing

Figure 3: The representative examples (reconstruction error $\ell_1 < 0.01$) between the observed $u(t + 1)$ and predicted $\hat{u}(t + 1)$ tau SUVRs generated by *MFG4AD* (left: CN, right: AD). Cognitively normal (CN), Alzheimer's disease (AD).

approaches. This superior performance stems from the integration of a *Wasserstein*-1 Lagrangian GAN, which improves the fidelity of synthesized tau patterns through distributional alignment in the prediction space, and a reaction–diffusion framework that explicitly captures the biophysical dynamics of tau propagation modulated by amyloid interaction. Representative predictions generated by *MFG4AD* are visualized in Fig. 3, where the absolute difference between observed and predicted

tau SUVR remains below 0.01, i.e., $\ell_1 = |u(t+1) - \hat{u}(t+1)| < 0.01$. Additional visualizations are provided in Appendix A.2. We further compared prediction errors (MAE and RMSE) across diagnostic groups in the ADNI and OASIS cohorts (Fig. 4a). In ADNI, both metrics are low, particularly in the EMCI and LMCI groups, with the lowest MAE observed in EMCI (0.0600 ± 0.0253) and the lowest RMSE in LMCI (0.0787 ± 0.0072). In contrast, OASIS exhibits higher errors in both CN and AD groups, especially in AD (RMSE = 0.7040 ± 0.130), likely reflecting greater cohort heterogeneity. Nevertheless, our method maintains the best overall predictive accuracy across all settings. To further assess spatial modeling fidelity, we visualized cortical maps of vertex-wise absolute prediction errors on ADNI and OASIS datasets (Fig. 4b). Both datasets exhibit localized discrepancies, with larger deviations concentrated in temporal and medial regions known for high tau variability. Overall, ADNI shows lower error magnitudes compared to OASIS, supporting the robustness of our model across heterogeneous populations. Finally, we evaluated model performance by comparing predicted and ground-truth mean tau SUVR values for each diagnostic subgroup (Fig. 4c). A simple linear regression was performed to quantify the agreement between predicted and observed values across subjects. In ADNI (left), strong linear relationships were observed across all disease stages, with a mean slope of 0.98 ± 0.06 and $R^2$ of 0.91 ± 0.07 (CN/SMC: 1.052 | 0.822; EMCI: 0.975 | 0.953; LMCI: 1.001 | 0.895; AD: 1.125 | 0.982). In OASIS (right), the mean slope was 0.90 ± 0.16 with $R^2$ of 0.660 ± 0.17 (CN: 1.007 | 0.785; AD: 0.784 | 0.535). These results indicate strong agreement between predicted and observed tau SUVR values, particularly in earlier disease stages and in the ADNI cohort, highlighting the stability and generalizability of our predictive framework.

Table 1: Prediction performance (MAE/RMSE) on ADNI and OASIS. '*' denotes the significant improvement ($p$-value <0.01, paried t-test.)

| Model | ADNI | | OASIS | | ADNI | | OASIS | |
|---|---|---|---|---|---|---|---|---|
| | MAE | RMSE | MAE | RMSE | MAE | RMSE | MAE | RMSE |
| DNN | $0.156^*_{\pm0.026}$ | $0.235^*_{\pm0.047}$ | $0.476^*_{\pm0.092}$ | $0.638^*_{\pm0.119}$ | $0.148^*_{\pm0.025}$ | $0.237^*_{\pm0.053}$ | $0.497^*_{\pm0.029}$ | $0.668^*_{\pm0.027}$ |
| GCN | $0.158^*_{\pm0.017}$ | $0.249^*_{\pm0.032}$ | $0.458^*_{\pm0.057}$ | $0.637^*_{\pm0.064}$ | $0.157^*_{\pm0.015}$ | $0.249^*_{\pm0.029}$ | $0.459^*_{\pm0.058}$ | $0.641^*_{\pm0.066}$ |
| GCNII | $0.178^*_{\pm0.032}$ | $0.299^*_{\pm0.068}$ | $0.488^*_{\pm0.055}$ | $0.666^*_{\pm0.082}$ | $0.127^*_{\pm0.008}$ | $0.189^*_{\pm0.015}$ | $0.469^*_{\pm0.060}$ | $0.649^*_{\pm0.087}$ |
| NDM | $0.103^*_{\pm0.021}$ | $0.135^*_{\pm0.029}$ | $0.484_{\pm0.212}$ | $0.613^*_{\pm0.258}$ | $0.101^*_{\pm0.018}$ | $0.1321^*_{\pm0.026}$ | $0.459_{\pm0.194}$ | $0.589^*_{\pm0.246}$ |
| Neuro-ODE | $0.127^*_{\pm0.014}$ | $0.190^*_{\pm0.020}$ | $0.485^*_{\pm0.048}$ | $0.655^*_{\pm0.069}$ | $0.127^*_{\pm0.008}$ | $0.189^*_{\pm0.015}$ | $0.487^*_{\pm0.054}$ | $0.664^*_{\pm0.076}$ |
| GRAND | $0.181^*_{\pm0.031}$ | $0.305^*_{\pm0.068}$ | $0.495^*_{\pm0.057}$ | $0.667^*_{\pm0.079}$ | $0.214^*_{\pm0.028}$ | $0.340^*_{\pm0.061}$ | $0.469^*_{\pm0.060}$ | $0.649^*_{\pm0.087}$ |
| GREAD | $0.163^*_{\pm0.021}$ | $0.269^*_{\pm0.051}$ | $0.488^*_{\pm0.0644}$ | $0.685^*_{\pm0.098}$ | $0.195^*_{\pm0.024}$ | $0.344^*_{\pm0.086}$ | $0.463^*_{\pm0.069}$ | $0.653^*_{\pm0.092}$ |
| Ridge | $0.090^*_{\pm0.010}$ | $0.132^*_{\pm0.024}$ | $0.456_{\pm0.053}$ | $0.620_{\pm0.062}$ | $0.088^*_{\pm0.012}$ | $0.132^*_{\pm0.027}$ | $0.451_{\pm0.051}$ | $0.623_{\pm0.063}$ |
| DSM | $0.087^*_{\pm0.010}$ | $0.163^*_{\pm0.025}$ | $0.468^*_{\pm0.061}$ | $0.727_{\pm0.073}$ | $0.083^*_{\pm0.004}$ | $0.129_{\pm0.019}$ | $0.447_{\pm0.037}$ | $0.702_{\pm0.068}$ |
| GAN | $0.247^*_{\pm0.019}$ | $0.343^*_{\pm0.046}$ | $0.490^*_{\pm0.051}$ | $0.663^*_{\pm0.070}$ | $0.255^*_{\pm0.021}$ | $0.361^*_{\pm0.044}$ | $0.517^*_{\pm0.071}$ | $0.694^*_{\pm0.098}$ |
| *MFG4AD* | — | — | — | — | $\mathbf{0.064_{\pm0.004}}$ | $\mathbf{0.093_{\pm0.009}}$ | $\mathbf{0.435_{\pm0.066}}$ | $\mathbf{0.619_{\pm0.061}}$ |
| Input | | Tau | | | | Tau+amyloid | | |

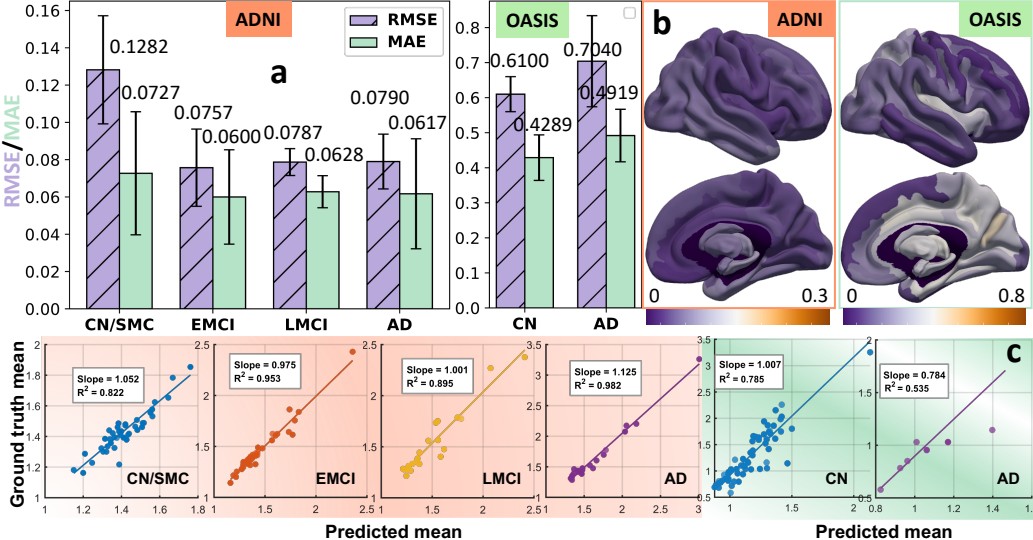

Figure 4: (a) Comparison of prediction errors across diagnostic subgroups in the ADNI (left) and OASIS (right) cohorts. (b) Vertex-wise absolute error maps show spatial patterns of prediction errors, with larger deviations in temporal and medial regions. (c) Predicted vs. observed mean tau SUVR values for each subgroup.

**Ablation Study.** To tease apart the roles of the two mechanistic terms in our model, we ablate the *diffusion* component ($S$) and *reaction* term ($R$), the results are summarized in Fig 5. Removing either component results in a noticeable degradation in performance, confirming that both processes are necessary for precise tau-propagation modelling. Eliminating the reaction term produces the larger error increase, underscoring amyloid burden as a key modulator of tau dynamics, whereas suppressing diffusion breaks the spatial continuity required to capture network-based spread. The learned scaling factors $\{\mu_1 = 0.94, \mu_2 = 0.97, \mu_3 = 1.33\}$ make this balance explicit: advection ($\mu_1$) and diffusion ($\mu_2$) contribute almost equally, while the amyloid–tau interaction term $\mu_3$ arries the greatest weight. This slightly elevated interaction coefficient, paired with near-unity transport coefficients, yields a physiologically plausible picture in which A$\beta$ "hot-spots" seed local tau build-up, then the advection–diffusion machinery conveys pathology throughout the connectome. Such dynamics mirror Braak staging [3] patterns and recent multimodal PET observations, and support the biological view that A$\beta$ deposition primes regions for accelerated tau spread once both pathologies co-localise, an effect repeatedly reported in experimental studies [20].

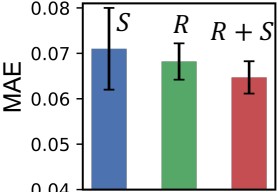

Figure 5: Ablation of diffusion ($S$) and reaction ($R$) components on performance.

## 3.2 Biologically-Informed Interpretation of Tau Propagation

**Tau Propagation Pathways on the Cortical Surface.** Fig. 6 illustrates the modeled evolution of tau pathology across different stages of AD in both the ADNI (top) and OASIS (bottom) datasets. For each group (CN, EMCI, LMCI, AD), we present the initial tau distribution $\rho_0$, the follow-up accumulation $\rho_1$, and the estimated propagation field $\nu$. The flow fields (last column of each group) reveal the direction and magnitude of tau propagation, where red colors indicate stronger spread flux. The color intensity in $\rho$ maps indicates the population-averaged tau accumulation, while the flow field $\nu$ captures the dominant direction and strength of tau transport across the cortical surface. A clear progression in tau flow strength is observed across disease stages: CN individuals exhibit minimal propagation, while tau flow becomes progressively more prominent in EMCI, LMCI, and reaches its peak in AD. This trend is consistent in both datasets and reflects the escalating spatial spread of tau pathology as the disease advances. Notably, the temporal lobe (indicated in red regions) shows strong and persistent involvement, serving as a key hub for tau diffusion in later stages. Importantly, these modeled propagation patterns align with established neuropathological findings, where tau pathology is known to originate in the transentorhinal and entorhinal cortex, before spreading to the hippocampus and neocortex in a stereotyped fashion [3]. This consistency with Braak [3] staging reinforces the biological plausibility of our model in capturing disease-relevant tau dynamics.

**New Insights into A$\beta$–Tau Interactions in AD.** To probe the mechanistic basis of A$\beta$–tau interactions, we analyzed the symbolic reaction functions $R_j(u, v)$ learned by our model, where each term encodes how amyloid burden $v_j$ at region $j$ contributes to tau accumulation at region $i$. By systematically scanning across all cortical vertices, we generated a spatial map (Fig. 7) in which darker shading highlights regions whose amyloid load most strongly drives downstream tau propagation. Two principal epicenters emerge: (1) *Medial Temporal Lobe (pink dashed region)* encompassing the entorhinal cortex and parahippocampal gyrus, the canonical nidus of early tauopathy (Braak I–II) [3]. Elevated amyloid levels in this region precipitate a cascade of tau spreading into neighboring isocortical territories. (2) *Medial Prefrontal Cortex (blue dashed region)* consistent with Thal A$\beta$ phases 1–2 [37], where surpassing an amyloid threshold triggers accelerated propagation from the medial temporal hub into anterior cortical areas, reinforcing the pathological cascade. To illustrate a representative symbolic reaction law, we selected a vertex in the `oc-temp_med-Parahip` region. Its update is given by $u(t+1) = -0.09*u_{44}+0.02*v_{81}-0.2*sin(2.2*v_9)+1.9+1.9/(\exp(3.5*v_{16})+1)$, where $u_{44}$ corresponds to the `S_calcarine` region, $v_{81}$ to `S_cingul-Mid-Ant`, $v_9$ to `G_cingul-Post-dorsal`, and $v_{16}$ to `G_front_sup-parahippocampal`. These interconnected nodes form the entorhinal–hippocampal–cingulate loop—a circuit widely recognized as the earliest site of tau accumulation in AD (Braak I–III). As a downstream hub of this loop, `oc-temp_med-Parahip` exhibits tau increase at $t + 1$ that is jointly driven by local A$\beta$–tau interactions and diffusion flux from upstream medial nodes. In contrast, remote regions such as `G_orbital` (orbital frontal cortex) typically become involved only in later Braak stages and do not participate in this early propagation network. For example, $-0.09u_{44}$: The negative coefficient from the calcarine cortex (a primary visual region affected in late Braak stages V–VI) may reflect a dampening influence on tau accumulation in early-stage regions—potentially capturing long-range regulatory effects in network dynamics. Constant term

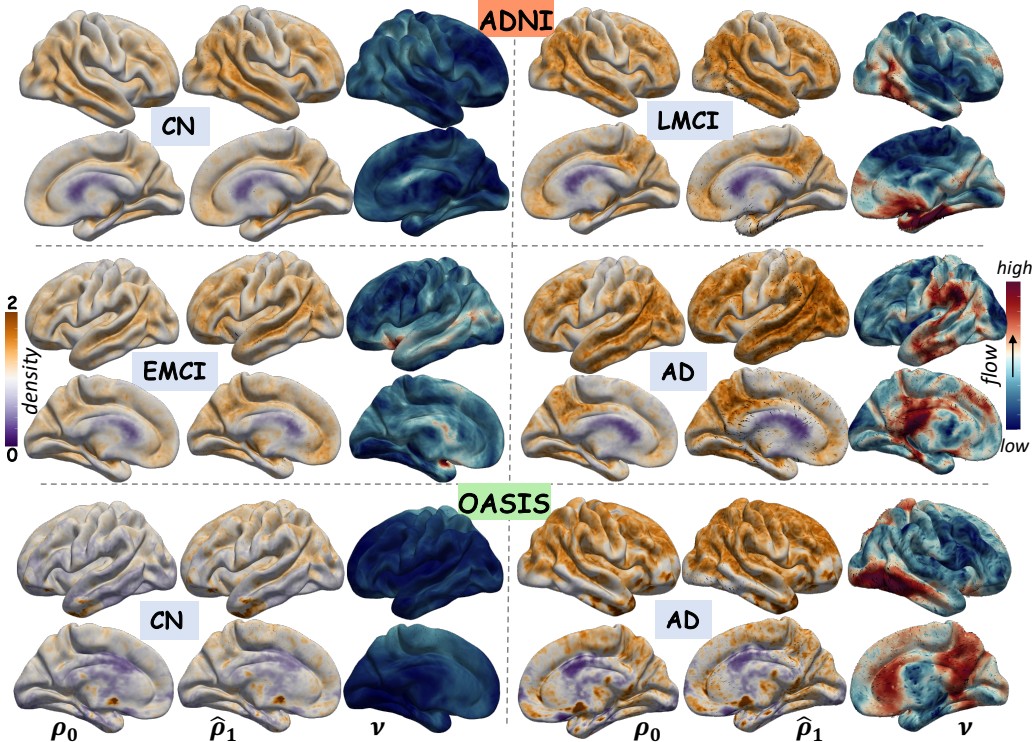

Figure 6: Visualization of tau propagation across stages of AD progression in ADNI (top) and OASIS (bottom).

(+1.9): Captures the intrinsic baseline accumulation of tau in the medial temporal lobe, consistent with spontaneous age-related tauopathy in entorhinal and parahippocampal areas [4]. Last term $1.9/(\exp(3.5 * v_{16}) + 1)$: A sigmoid-shaped A$\beta$ term, indicative of a saturating dose-response, reflects known dynamics where tau is more sensitive to A$\beta$ at subthreshold levels but less responsive once amyloid burden becomes extensive [24]. Taking together, these results show that *MFG4AD* learns biologically meaningful reaction laws, linking local A$\beta$ burden and network connectivity to tau spread [33]. The derived symbolic equations reveal early epicenters and propagation trajectories consistent with Braak I–III staging, offering interpretable and predictive insights into AD progression.

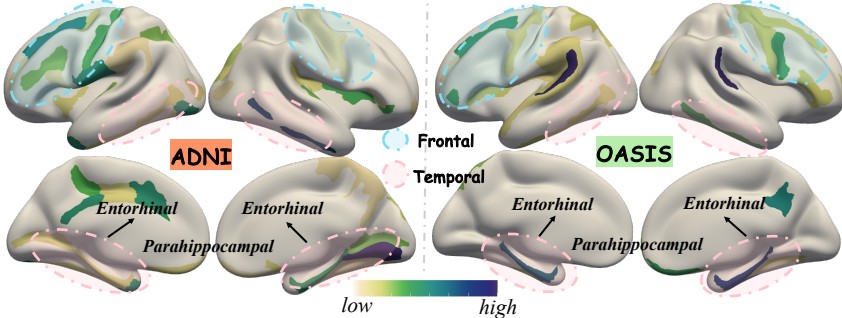

Figure 7: The brain surface mapping of A$\beta$-tau interaction. Dark color indicates active involvement of amyloid cascade in the tau propagation.

## 4   Conclusion

In this work, we introduced *MFG4AD*, a unified, physics-informed deep learning framework that: (1) models tau spread as a network-constrained reaction–diffusion process with a data-driven symbolic law for tau–amyloid crosstalk; (2) casts this system as an equivalent potential mean field game, linking classical PDE theory to tau propagation; and (3) employs a *Wasserstein*-1 Lagrangian GAN to learn optimal transport flows for accurate tau forecasting. On ADNI and OASIS cohorts, *MFG4AD* delivers state-of-the-art predictions for unseen subjects and resolves tau-flow directions, pinpointing peak-flux hotspots, while also uncovering an explicit, interpretable reaction law, offering a powerful combination of predictive performance and mechanistic insight into Alzheimer's pathology.

## Acknowledgement

This work was supported by the National Institutes of Health (AG091653, AG068399, AG084375) and the Foundation of Hope. The views and conclusions contained in this document are those of the authors and should not be interpreted as representing the official policies, either expressed or implied, of the NIH.

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

# A Technical Appendices and Supplementary Material

## A.1 Brain Network Construction from Cortical Surface Mesh

To capture anatomically faithful pathways for tau propagation along the cortical mantle, we constructed a sparse, geometry-aware graph based on the native cortical surface topology. Specifically, we utilized the `lh/rh.pial` surface mesh of the left/right hemisphere generated by FreeSurfer [13], which represents the cortical sheet as a triangular mesh composed of $N = 163,842$ vertices and approximately 327,680 faces.

Each triangular face defines three local connections between mesh vertices. We constructed a graph $\mathcal{G} = (\mathcal{X}, \mathcal{E}, D)$ by treating each vertex as a node $x_i \in \mathcal{X}$, and adding an undirected edge $(x_i, x_j) \in \mathcal{E}$ if the vertices $x_i$ and $x_j$ are connected by at least one triangle. This results in a topology-preserving adjacency matrix $D \in \mathbb{R}^{N \times N}$ encoding binary connectivity that reflects local anatomical continuity.

To integrate geometric information relevant to spatial diffusion, we assigned edge weights based on the Euclidean distance between connected vertices. For each edge $(x_i, x_j)$, the weight was defined as:

$$D_{ij} = \left\| x_i - x_j \right\|_2 = \sqrt{(a_i - a_j)^2 + (b_i - b_j)^2 + (c_i - c_j)^2} \tag{13}$$

where $x_i = (a_i, b_i, c_i), x_j = (a_j, b_j, c_j) \in \mathbb{R}^3$ denote the 3D coordinates of vertices $x_i$ and $x_j$ (Note, we use $x_{i,j}$ to represent the index and coordinates of the node uniformly for simplicity). The resulting weighted adjacency matrix provides a biologically plausible scaffold for modeling local propagation dynamics constrained to the cortical surface, the illustration is shown in Fig. 8.

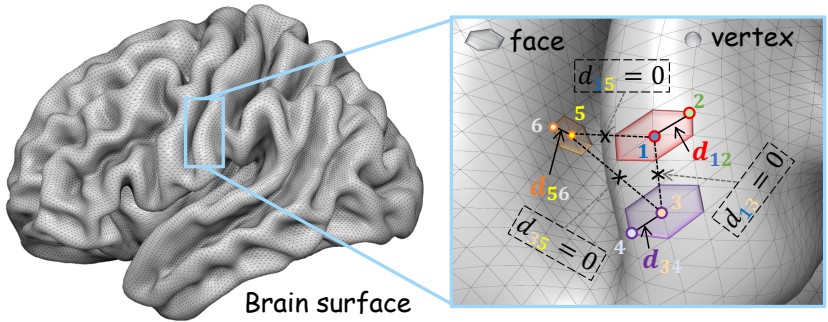

Figure 8: An illustration of constructing brain network topology.

The constructed graph exhibits an average node degree of approximately 6, consistent with the local connectivity induced by the triangular tessellation of the cortical sheet. In contrast to conventional $k$-nearest-neighbor ($k$NN) graphs that are built solely based on Euclidean proximity, this approach ensures geometric and topological consistency by avoiding spurious long-range connections that may cross sulcal boundaries or violate the anatomical folding patterns.

To support downstream spectral and learning-based analyses, we further derived the symmetrically normalized graph Laplacian: $L_{\text{sym}} = I - A^{-1/2} D A^{-1/2}$, where $A$ is the diagonal degree matrix with $A_{ii} = \sum_j D_{ij}$.

Finally, for compatibility with graph learning frameworks such as PyTorch Geometric, we exported the graph as an edge list `edge_index` $\in \mathbb{Z}^{2 \times |\mathcal{E}|}$ and a corresponding edge weight vector `edge_weight` $\in \mathbb{R}^{|\mathcal{E}|}$.

Our approach leverages the native triangular mesh of the cortical surface, where each node is connected to its immediate neighbors based on surface topology. This results in a fixed and biologically grounded neighborhood structure, typically with an average node degree of 6. The mesh graph strictly adheres to the geometry of the cortical sheet, preserving anatomical continuity and avoiding non-local shortcuts.

This anatomically faithful structure is especially important for modeling prion-like tau propagation, which is believed to follow trans-neuronal transmission along physically connected pathways. Clava-

guera et al. [12], for example, demonstrated that tau pathology spreads from the injection site to anatomically connected regions, supporting the need for realistic graph representations that reflect underlying biological constraints.

## A.2 Implementation Details and Experimental Results

The pseudocode for our method is presented in Algorithm 1. The full implementation—including all hyperparameter settings—is available from our anonymous GitHub repository: `https://github.com/Dandy5721/MFG4AD2025`.

---

**Algorithm 1** Training *MFG4AD* for Tau Dynamics

---

**Require:** For each subject, $u(t), u(t+1) \in \mathbb{R}^N$ (tau at times $t$ and $t+1$) $\quad v(t) \in \mathbb{R}^N$ (amyloid at time $t$), the coordinates of each vertex $x_i \subset \mathcal{X}$, learning rates $\eta_C, \eta_G > 0$, clip threshold $b > 0$, critic steps $n_C$, the weight of reconstruction $\lambda$

1: **while** not converged **do**
2: $\quad$ Build graph $\mathcal{G} = (X, D)$ with distances $d_{ij} = \|x_i - x_j\|$.
3: $\quad$ **for** $k = 1, \dots, n_C$ **do**
4: $\quad\quad$ // — CRITIC UPDATE —
5: $\quad\quad$ Obtain flow field $\nu \in \mathbb{R}^{N \times d}$ by $G_\theta$
6: $\quad\quad$ Predict $\hat{u}(t+1)$ via Eq. (12)
7: $\quad\quad$ Define the loss of *Critic*: $\mathcal{L}_C \leftarrow \frac{1}{N} \sum_{i=1}^N C_\varphi\big(\hat{u}_i(t+1)\big) - \frac{1}{N} \sum_{i=1}^N C_\varphi\big(u_i(t+1)\big)$
8: $\quad\quad$ Gradient-ascent step $\varphi \leftarrow \varphi + \eta_C \nabla_\varphi \mathcal{L}_C$ $\quad\quad\triangleright$ Spectral Norm enforces 1-Lipschitz
9: $\quad$ **end for**
10: $\quad$ // — GENERATOR UPDATE —
11: $\quad$ $\nu \leftarrow G_\theta\big(u(t), v(t)\big)$ $\quad\quad\quad\quad\quad\quad\quad\quad\triangleright$ $G_\theta$ is composed of $S_\varepsilon$, $R_\xi$, and $H_\vartheta$
12: $\quad$ $\hat{u}(t+1) \leftarrow$ Eq. 12
13: $\quad$ *Generator* loss: $\mathcal{L}_G \leftarrow -\frac{1}{N} \sum_{i=1}^N C_\varphi\big(\hat{u}_i(t+1)\big) + \lambda \frac{1}{N} \sum_{i=1}^N |\hat{u}_i(t+1) - u_i(t+1)|_1$
14: $\quad$ Gradient-descent step $\theta \leftarrow \theta - \eta_G \nabla_\theta \mathcal{L}_G$
15: **end while**

---

More visualization results generated by our proposed *MFG4AD* on ADNI and OASIS datasets in Fig. 9.

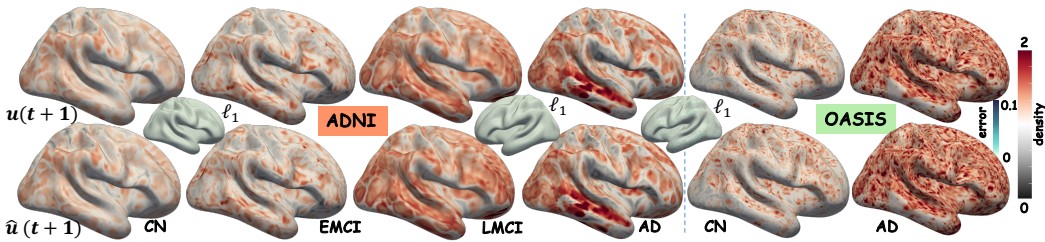

Figure 9: The representative examples (reconstruction error $\ell_1 < 0.02$) between the observed $u(t+1)$ and predicted $\hat{u}(t+1)$ tau SUVRs generated by *MFG4AD* (left: ADNI, right: OASIS). Cognitively normal (CN), early mild cognitive impairment (EMCI), late mild cognitive impairment (LMCI).

All experiments were conducted on an RTX A5000 GPU. The corresponding inference times are reported in Table 2. The main cost of NMD is the diffusion kernel $e^{-\beta L t}$ which involves matrix exponential operation.

Table 2: The inference time for each model.

| Model | DNN | GCN | GCNII | NDM | Neuro-ODE | - |
|-------|-----|-----|-------|-----|-----------|---|
| Time(s) | 0.05 | 0.09 | 0.18 | 0.63 | 0.04 | - |
| Model | GRAND | GREAD | DSM | GAN | Ridge | *MFG4AD* |
| Time (s) | 0.14 | 0.36 | 0.26 | 0.05 | 0.01 | 0.27 |

## A.3    Discussion and Limitation

As a proof-of-concept, we leverage the analytic reaction–diffusion laws discovered by our model to ask a fundamental question: Does amyloid drive tau aggregation locally within the same region, or remotely across distinct cortical areas? By fitting symbolic expressions at every vertex, we observe a hybrid interaction: amyloid deposits both amplify tau buildup in their own region and "prime" downstream nodes for accelerated spread. In addition, our current framework fits an independent reaction law at each of the  100K surface vertices, which greatly increases computation and memory costs. To address this, we distill these per-vertex laws into a single, regionally parameterized global reaction function defined over cortical subdomains of  1,000 vertices each—preserving interpretability while enabling fast, large-scale prediction.

The surface-based mesh graph constructed from cortical triangular tessellation provides an anatomically faithful substrate for modeling prion-like tau propagation along the cortical mantle. This structure is especially suited for simulating local trans-neuronal spread that adheres to physical cortical continuity, which characterizes the early stages of pathological tau aggregation. To capture long-range propagation, future extensions may incorporate structural connectivity data (e.g., tractography-based inter-regional projections).

In the future, we will cross-validate our framework on additional AD and AD-related cohorts, extend the symbolic module to capture interactions with other biomarkers (e.g. neuroinflammation, synaptic loss), and perform disease simulations driven by our reaction–diffusion engine to test hypothetical interventions before clinical trials.

## A.4    Impact Statement

From a machine-learning perspective, our work introduces a physics-informed *Wasserstein* Lagrangian GAN combined with symbolic regression to learn interpretable, PDE-like reaction–diffusion dynamics directly on irregular cortical graphs, bridging black-box GNNs and white-box biophysical models and yielding reusable "reaction–diffusion engines" for spatiotemporal forecasting. From a neuroscience standpoint, the same framework uncovers explicit amyloid–tau interaction kernels and cortical propagation pathways, quantitatively reproducing tau-spread patterns consistent with Braak staging and pinpointing vulnerable hub regions whose amyloid burden drives downstream aggregation, thereby providing a data-driven foundation for mechanistic hypothesis testing and targeted intervention in Alzheimer's disease. Ultimately, by translating these mechanistic insights into personalized predictive tools, our approach paves the way for earlier diagnosis and more effective, tailored therapies in clinical practice.

