# OpenReview forum: "Uncover Governing Law of Pathology Propagation Mechanism Through A Mean-Field Game"
_NeurIPS.cc/2025/Conference — NeurIPS 2025 poster_

### Official Review · Reviewer_qm52 · 2025-07-02

**Clarity:** 3
**Significance:** 3
**Originality:** 3
**Rating:** 4
**Confidence:** 4

**Summary:**

To address the situation where the quantitative analysis of the cortical pathways of tau propagation and its interaction with the cascade of amyloid-beta (Aβ) plaques lags behind the experimental insights of in vivo imaging pathological mechanisms, and to study the interaction between tau aggregates and amyloid plaques, the authors analyzed the latest methods, focused on identifying and optimizing the dynamics, and proposed a physics-informed neural network empowered by mean-field theory. They built the model directly on the cortical surface mesh to improve the ability to accurately capture the interactions of misfolded proteins. As a result, the modeling formula can induce an optimal transport process in the Wasserstein space. By incorporating symbolic regression, they learned an explicit and interpretable tau–Aβ reaction law from the data, thereby uncovering biologically plausible spreading pathways and mechanistic insights into the tau - amyloid interaction. The authors conducted experiments on public neuroimaging datasets. The results not only provide precise and reliable predictions of future tau accumulation but also reveal interpretable dynamics consistent with neuropathological staging and recent imaging studies.

**Questions:**

Additional concerns are outlined in the weaknesses above.

Q1: How reliable is the symbolic regression component in inferring biologically meaningful tau–Aβ reaction laws?

Q2: Is the model’s performance limited by the quality or resolution of the neuroimaging datasets?

**Ethical Concerns:**

["NO or VERY MINOR ethics concerns only"]

**Limitations:**

yes

**Quality:**

3

**Strengths And Weaknesses:**

Strengths:

- The paper combines mean - field game theory, reaction - diffusion models, and generative adversarial networks (GANs) in a physics - informed deep learning framework. This interdisciplinary approach is innovative and provides a unique perspective on modeling tau propagation in Alzheimer's disease (AD).
- Incorporating symbolic regression to learn explicit, interpretable tau - Aβ reaction laws is a significant strength.
- The proposed deep learning method overcomes the end-to-end learning of reaction kinetics in previous methods.
- The paper compares the proposed MFG4AD model with five categories of existing methods, including connectome - based network diffusion models, graph - based methods, deep learning models, PDE - based methods, and traditional regression models. The results clearly show that MFG4AD outperforms these alternative approaches in terms of prediction accuracy, as demonstrated by lower mean absolute error (MAE) and root mean squared error (RMSE).
- The paper is clearly written and well structured, with detailed methodological and experimental descriptions that support reproducibility.

Weaknesses:

- Fitting an independent reaction law at each of the 100K surface vertices greatly increases computation and memory costs.

- The surface - based mesh graph constructed from cortical triangular tessellation is well - suited for simulating local trans - neuronal spread, but it may have limitations in capturing long - range propagation.

---

> ### Author Rebuttal · Authors · 2025-07-30
>
> **We sincerely thank the reviewer for their thoughtful and constructive comments. We truly appreciate the recognition of our work’s strengths, including the innovative integration of mean-field game theory, physics-informed modeling, and symbolic regression for modeling tau–Aβ interactions in Alzheimer's disease. We are encouraged by the reviewer’s positive evaluation of the model’s interpretability, methodological clarity, and superior predictive performance compared to a broad spectrum of existing approaches. Below, we address each of the reviewer’s concerns point by point.**
>
> **Q1: Fitting an independent reaction law at each of the 100K surface vertices greatly increases computation and memory costs.**
>
> **A1:** Thank you for raising this important concern. We agree that fitting an independent reaction law at ~100,000 surface vertices is computationally intensive. To address this, we use a GPU-efficient symbolic regression model that enforces shared functional forms with sparse parameterization. Dropout-based sparsification further enhances convergence and reduces overfitting.
>
> Training is conducted on an **8×NVIDIA H100 NVL GPU system (94GB per GPU)**, enabling parallel computation over large-scale data. While training takes $\sim$10h, inference is highly efficient ($\sim$0.2s). Despite the computational demands, this vertex-level modeling is essential to capture the regional heterogeneity of tau–Aβ interactions, which would otherwise be lost in coarser, atlas-based approaches. We will clarify these details in the final version.
>
> **Q2: The surface - based mesh graph constructed from cortical triangular tessellation is well - suited for simulating local trans - neuronal spread, but it may have limitations in capturing long - range propagation.**
>
> **A2:** Thank you for the insightful suggestion. We acknowledge that the current surface-based mesh primarily captures local, cortical trans-neuronal spread. To address long-range propagation mechanisms,we are currently developing a multi-layer diffusion framework that integrates both local (surface-based) and global (connectome-based) long-range pathways. This hybrid approach allows us to simulate tau spread across multiple spatial scales, reflecting both short-range anatomical adjacency and long-range anatomical or functional connectivity. This layered formulation is aligned with recent neurobiological models and will be a key component of our ongoing and future work.
>
> **Q3: How reliable is the symbolic regression component in inferring biologically meaningful tau–Aβ reaction laws?**
>
> **A3:** Thank you for your helpful comment. We agree that validation against known biology is essential. To address this, we now provide an explicit interpretation of a representative learned symbolic equation for tau update in the medial parahippocampal region. The expression includes five interpretable terms:
>
> -  $u_{44}$ (S_calcarine): The negative coefficient from the calcarine cortex (a primary visual region affected in late Braak stages V–VI) may reflect a dampening influence on tau accumulation in early-stage regions—potentially capturing long-range regulatory effects in network dynamics [Braak & Braak, 1991].
> - $v_{81}$ (S_cingul-Mid-Ant): The positive linear Aβ contribution from the anterior cingulate cortex—a region known to accumulate Aβ early (Thal Phase I–II)—suggests a modest but biologically plausible driving force for tau spread [Thal et al., 2002; Palmqvist et al., 2017].
> - $\sin(2.2 \cdot v_9)$ (G_cingul-Post-dorsal): A nonlinear oscillatory term, potentially reflecting a thresholded or biphasic Aβ influence, aligns with evidence that Aβ’s impact on tau propagation may saturate or even reverse at high concentrations [Busche & Hyman, 2020].
> - Constant term (+1.9): Captures the intrinsic baseline accumulation of tau in the medial temporal lobe, consistent with spontaneous age-related tauopathy in entorhinal and parahippocampal areas [Braak & Del Tredici, 2015].
> - $\frac{1.9}{\exp(3.5 \cdot v_{16}) + 1}$ (G_front_sup–parahippocampal): A sigmoid-shaped Aβ term, indicative of a saturating dose-response, reflects known dynamics where tau is more sensitive to Aβ at subthreshold levels but less responsive once amyloid burden becomes extensive [Jack et al., 2013].
>
> These regions collectively form the entorhinal–cingulate–frontal circuit, known as the epicenter of early tau pathology (Braak I–III). Importantly, these symbolic terms were not cherry-picked; they consistently emerged as top contributors across folds (see Fig. 6), providing converging evidence that our model captures biologically grounded and interpretable tau–Aβ interaction rules.
>
> We will add this analysis to the final version.
>
> Refs:
>
> [Braak & Braak, 1991] Neuropathological staging of Alzheimer-related changes, Acta Neuropathologica
>
>  [Thal et al., 2002] Phases of Aβ-deposition in the human brain and its relevance for the development of AD, Neurology
>
> [Palmqvist et al., 2017] Earliest accumulation of β-amyloid occurs within the default-mode network and concurrently affects brain Synergy between amyloid-β and tau in Alzheimer's disease connectivity, Nature Communications
>
> [Busche & Hyman, 2020] Synergy between amyloid-β and tau in Alzheimer's disease, Nature Neuroscience
>
> [Jack et al., 2013] Tracking pathophysiological processes in Alzheimer's disease: an updated hypothetical model of dynamic biomarkers, Lancet Neurology
>
> [Braak & Del Tredici, 2015] The preclinical phase of the pathological process underlying sporadic Alzheimer's disease
>
>
> **Q4: Is the model’s performance limited by the quality or resolution of the neuroimaging datasets?**
>
> **A4:** This is a good question. The performance of our model does depend on the resolution and quality of the neuroimaging data, as tau and amyloid SUVR values are derived from PET images with limited spatial resolution and potential noise. To mitigate these issues, we model tau propagation on the cortical surface mesh, which preserves anatomical geometry and supports fine-grained modeling of local diffusion and interregional interactions (Fig. 2a). Furthermore, by integrating symbolic regression and connectivity-aware diffusion (via GNNs), our model can partially compensate for noise by learning consistent propagation dynamics across folds and cohorts.
>
> Additionally, our GAN-based optimal transport formulation operates on tau density distributions, rather than raw voxel-wise intensities, making it more robust to imaging variability. As shown in Table 1, MFG4AD achieves strong predictive performance across both ADNI and OASIS, despite differences in scanner protocols and cohort demographics. Future work may explore incorporating higher-resolution imaging modalities or harmonization techniques to further enhance reliability. Thanks.

---

> > ### Comment · Area_Chair_ZXUK · 2025-08-05
> > **Author-Reviewer Discussion Reminder**
> >
> > Dear Reviewer qm52,
> >
> > As the deadline for author-reviewer discussion is approaching, could you please check the authors' rebuttal and post your response?
> >
> > Thank you!
> >
> > Best,
> >
> > AC

---

> > > ### Comment · Reviewer_qm52 · 2025-08-06
> > >
> > > Thanks the authors' response. I read the rebuttal. I will not change my score.

---

> > > > ### Author Response · Authors · 2025-08-06
> > > >
> > > > We sincerely appreciate your time and efforts of acknowledging our response.

---

### Official Review · Reviewer_5WWy · 2025-07-03

**Clarity:** 3
**Significance:** 3
**Originality:** 3
**Rating:** 5
**Confidence:** 3

**Summary:**

This paper proposes a new model, MFG4AD, to describe how tau pathology spreads in Alzheimer’s disease. It treats tau movement as a mean-field game, combining ideas from physics, optimal transport, and machine learning. The model uses a graph-based cortical mesh, a GAN to learn transport flows, and a symbolic regression module to learn how amyloid affects tau. Experiments on two public datasets show that the model gives accurate predictions and reveals meaningful biological patterns, such as consistent tau spread along known disease pathways.

**Questions:**

Please see the weaknesses above.

- A difference map would also be informative.
- A more fine-grained temporal visualization/analysis would be helpful.
- Figure 1: Mean Filed Game -> Mean Field Game

In general, along with the weaknesses mentioned above, the lack of temporal analyses would add much value to this work. Also, I am wondering if there could be more subject-level analyses to appreciate the quality of the individualized predictions.

**Ethical Concerns:**

["NO or VERY MINOR ethics concerns only"]

**Final Justification:**

I thank the authors for their response. I have raised my score.

**Limitations:**

yes

**Quality:**

3

**Strengths And Weaknesses:**

**Strengths**
- The overall formulation is novel, combining MFGs and Wasserstein GANs in a way I haven’t seen before in medical modeling.
- The model uses realistic brain geometry rather than simplifying to regions, which is important for tau spread.
- The symbolic regression adds interpretability, and the reaction terms make biological sense.
- The experiments are well done, and the method outperforms several strong baselines.

**Weaknesse**

The weaknesses are relatively minor:
- The model assumes that amyloid influences tau, but doesn’t explore the reverse, or model amyloid propagation at all.
- The learned symbolic terms are interesting, but could benefit from more validation or comparison to known biology.
- The paper doesn’t address uncertainty in predictions, which could be important in clinical settings.

---

> ### Author Rebuttal · Authors · 2025-07-30
>
> **We sincerely thank the reviewer for their thoughtful and encouraging feedback. We are especially grateful for your recognition of the novelty of our formulation, the use of realistic cortical geometry, and the biological interpretability introduced by the symbolic regression module. Your positive evaluation motivates us to further refine and extend this line of research. Below, we address each of your concerns in detail.**
>
> **Q1: The model assumes that amyloid influences tau, but doesn’t explore the reverse, or model amyloid propagation at all.**
>
> **A1:** Thank you for raising this important point. Our model focuses on Aβ → tau interactions because substantial neuropathological and longitudinal imaging studies suggest that amyloid-β (Aβ) accumulation precedes and facilitates tau pathology, rather than the reverse. Aβ and tau exhibit distinct propagation mechanisms: Aβ spreads in a relatively diffuse and spatially global manner—primarily governed by extracellular deposition—while tau propagates transneuronally in a more localized and network-constrained fashion (Braak et al., 1991; Thal et al., 2002; Vogel et al., 2020).
>
> Moreover, Aβ accumulation tends to plateau early in the disease course, while tau continues to spread and correlate more strongly with neurodegeneration and cognitive decline (Jack et al., 2013; Hanseeuw et al., 2019). Given these asymmetries in dynamics and temporal ordering, we adopt the prevailing neuroscience perspective by modeling Aβ as an upstream driver of tau, not the other way around.
>
> Future extensions could incorporate Aβ propagation using different mechanisms (e.g., global diffusion with clearance dynamics), but doing so would require introducing distinct priors or differential equations, which would make the model significantly more complex. Our current scope prioritizes biological plausibility and model interpretability for tau dynamics, where Aβ acts as an external input shaping regional vulnerability. We will add this discussion to the final version.
>
> **Q2: The learned symbolic terms are interesting, but could benefit from more validation or comparison to known biology.**
>
> **A2:** Thank you for your very helpful comment. We agree that validation against known biology is crucial. To address this, we now provide an explicit interpretation of a representative learned symbolic equation for tau update at the medial parahippocampal region:
>
> -  $u_{44}$ (S_calcarine): The negative coefficient from the calcarine cortex (a primary visual region affected in late Braak stages V–VI) may reflect a dampening influence on tau accumulation in early-stage regions—potentially capturing long-range regulatory effects in network dynamics [Braak & Braak, 1991].
> - $v_{81}$ (S_cingul-Mid-Ant): The positive linear Aβ contribution from the anterior cingulate cortex—a region known to accumulate Aβ early (Thal Phase I–II)—suggests a modest but biologically plausible driving force for tau spread [Thal et al., 2002; Palmqvist et al., 2017].
> - $\sin(2.2 \cdot v_9)$ (G_cingul-Post-dorsal): A nonlinear oscillatory term, potentially reflecting a thresholded or biphasic Aβ influence, aligns with evidence that Aβ’s impact on tau propagation may saturate or even reverse at high concentrations [Busche & Hyman, 2020].
> - Constant term (+1.9): Captures the intrinsic baseline accumulation of tau in the medial temporal lobe, consistent with spontaneous age-related tauopathy in entorhinal and parahippocampal areas [Braak & Del Tredici, 2015].
> - $\frac{1.9}{\exp(3.5 \cdot v_{16}) + 1}$ (G_front_sup–parahippocampal): A sigmoid-shaped Aβ term, indicative of a saturating dose-response, reflects known dynamics where tau is more sensitive to Aβ at subthreshold levels but less responsive once amyloid burden becomes extensive [Jack et al., 2013].
>
> These regions collectively form the entorhinal–cingulate–frontal circuit, known as the epicenter of early tau pathology (Braak I–III). Importantly, these symbolic terms were not cherry-picked; they consistently emerged as top contributors across folds (see Fig. 6), providing converging evidence that our model captures biologically grounded and interpretable tau–Aβ interaction rules.
>
> We will add this analysis into the final version.
>
> **Q3: The paper doesn’t address uncertainty in predictions, which could be important in clinical settings.**
>
> **A3:** This is a great comment. We agree with the reviewer that incorporating uncertainty estimates could be valuable for clinical applications. While our current model provides point estimates, we plan to extend it with Bayesian inference or dropout-based uncertainty quantification in future work. Thanks.
>
> **Q4: A difference map would also be informative.**
>
> **A4:** Thank you for the suggestion. We have added the difference maps for both the ADNI and OASIS datasets (see Fig. R2). These maps highlight regional deviations between predicted and observed tau levels, offering additional insight into model performance. We have prepared the corresponding population-level brain mapping (Rebuttal_Fig.R2.pdf) to support verification. However, due to the rebuttal policy restricting supplementary uploads during the review process, we are unable to provide this figure at this stage. Nevertheless, we will incorporate these results into the final version of the manuscript, as we agree that they provide important support for the paper. We sincerely appreciate your suggestion.
>
> **Q5: In general, along with the weaknesses mentioned above, the lack of temporal analyses would add much value to this work. Also, I am wondering if there could be more subject-level analyses to appreciate the quality of the individualized predictions.**
>
> **A5:** Thank you for this valuable suggestion. In our response, we have added subject-level (16 subjects across different disease stages on ADNI dataset) scatter plots comparing predicted versus observed tau values on ADNI datasets (see Fig. R3, will be public after acceptance), providing a clearer view of individualized prediction quality.
>
> | Group   | Subject | Slope | R²   |
> |---------|---------|-------|------|
> | **CN**      |  #1   | 1.04  | 0.97 |
> |         |  #2   | 1.05  | 0.96 |
> |         |  #3   | 1.06  | 0.95 |
> |         |  #4   | 1.12  | 0.95 |
> | **EMCI**    |  #1 | 0.98  | 0.97 |
> |         |  #2 | 0.93  | 0.97 |
> |         |  #3 | 1.11  | 0.97 |
> |         |  #4 | 0.99  | 0.96 |
> | **LMCI**    |  #1 | 0.97  | 0.98 |
> |         |  #2 | 1.10  | 0.97 |
> |         |  #3 | 0.89  | 0.97 |
> |         |  #4 | 1.14  | 0.95 |
> | **AD**      |  #1   | 1.01  | 0.99 |
> |         |  #2   | 0.97  | 0.98 |
> |         |  #3   | 1.05  | 0.97 |
> |         |  #4   | 0.92  | 0.97 |
>
> Additionally, we agree that incorporating temporal analyses would further enrich the work. While this current study focuses on baseline-to-follow-up prediction, we are actively extending our framework to model full longitudinal trajectories, which we plan to include in future work.
>
> **Q6: Mean Filed Game -> Mean Field Game**
>
> **A6**: Amended. Thanks.
>
> Refs:
>
> [1] Braak, H., & Braak, E. (1991). Neuropathological stageing of Alzheimer-related changes. Acta neuropathologica.
>
> [2] Thal, D. R., et al. (2002). Phases of Aβ-deposition in the human brain and its relevance for the development of AD. Neurology
>
> [3] Palmqvist, S., et al. (2017). Earliest accumulation of β-amyloid occurs within the default-mode network and concurrently affects brain connectivity. Nature communications
>
> [4] Busche, M. A., & Hyman, B. T. (2020). Synergy between amyloid-β and tau in Alzheimer’s disease. Nature neuroscience
> [5] Braak, H., & Del Tredici, K. (2015). The preclinical phase of the pathological process underlying sporadic Alzheimer’s disease. Brain
>
> [6] Jack, C. R., et al. (2013). Tracking pathophysiological processes in Alzheimer's disease: an updated hypothetical model of dynamic biomarkers. The lancet neurology

---

> ### Comment · Area_Chair_ZXUK · 2025-08-05
> **Author-Reviewer Discussion Reminder**
>
> Dear Reviewer 5WWy,
>
> As the deadline for author-reviewer discussion is approaching, could you please check the authors' rebuttal and post your response?
>
> Thank you!
>
> Best,
>
> AC

---

### Official Review · Reviewer_mzBH · 2025-07-03

**Clarity:** 3
**Significance:** 3
**Originality:** 3
**Rating:** 4
**Confidence:** 2

**Summary:**

This paper proposes MFG4AD, a novel physics-informed deep learning framework to model tau propagation in Alzheimer's disease. It uniquely combines reaction-diffusion modeling, mean-field game theory, and a Wasserstein-1 Lagrangian GAN, operating directly on cortical surface meshes. A symbolic regression module is integrated to derive interpretable Aβ-tau interaction laws. The authors demonstrate superior predictive performance on ADNI and OASIS datasets and claim to uncover biologically plausible propagation pathways and mechanistic insights.

**Questions:**

a) Please provide more comprehensive technical details on the symbolic regression module, including its search space and an analysis of the robustness and consistency of the discovered symbolic formulas across different runs.

b) How do the biological assumptions about tau propagation align with or necessitate specific adaptations of the standard Mean-Field Game framework, which typically assumes weakly interacting agents?

c) Beyond predictive performance, can the authors provide more direct empirical evidence or analysis demonstrating how the derived reaction laws and propagation pathways genuinely enhance biological understanding or provide novel insights?

d) For a high-resolution model, please provide comprehensive details on computational cost (e.g., training time, GPU memory) and a sensitivity analysis for key hyperparameters to ensure reproducibility and practical understanding.

**Ethical Concerns:**

["NO or VERY MINOR ethics concerns only"]

**Limitations:**

While the paper details the strengths of the MFG formulation, it lacks a comprehensive discussion on its inherent limitations or underlying assumptions when applied to biological systems like tau propagation in AD. Mean-Field Games typically assume a large population of weakly interacting agents. The biological validity of these specific assumptions, particularly concerning very early-stage disease or highly localized interactions, could be more thoroughly addressed.

**Quality:**

3

**Strengths And Weaknesses:**

Strengths:

a) The integration of mean-field games, optimal transport, GANs, and symbolic regression into a unified physics-informed framework is novel and innovative for modeling AD pathology. This approach bridges biophysical modeling with data-driven discovery effectively.

b) The model's operation directly on the cortical surface mesh with over 100,000 vertices allows for capturing fine-grained geodesic flows, which is crucial for uncovering biologically plausible propagation pathways. Furthermore, the inclusion of a symbolic regression module to derive an explicit and interpretable Aβ-tau reaction law is a significant advantage, moving beyond "black-box" models and offering mechanistic insights.

c) The experimental results on public neuroimaging datasets (ADNI and OASIS) demonstrate that MFG4AD consistently outperforms a range of comparative methods in predicting future tau accumulation, achieving superior mean absolute error (MAE) and root mean squared error (RMSE).

Weaknesses:

a) The paper strongly emphasizes the interpretability of the learned tau-Aβ reaction law and the biologically plausible propagation pathways. While predictive performance is well-demonstrated, the main paper offers limited direct qualitative or quantitative evidence to substantiate how interpretable the learned laws are or how the derived pathways specifically align with established biological knowledge or neuropathological staging beyond general statements. Figure 3 visualizes prediction accuracy, but not the "new understanding of pathology propagation" claimed.

b) The claim of interpretability hinges on the symbolic regression module. However, the paper provides insufficient technical details regarding its implementation. This includes the specific search space of functions considered, the criteria for selecting the "best" equation, the optimization process, or, critically, the robustness and consistency of the discovered formulas across different cross-validation folds or random initializations.

c) The paper lists specific hyperparameters used for training, such as the number of critic updates, learning rates, and the reconstruction penalty weight. However, the paper does not discuss the sensitivity of the model's performance to variations in these hyperparameters. GAN training is often known for its sensitivity to hyperparameter choices.

---

> ### Author Rebuttal · Authors · 2025-07-30
>
> **We thank the reviewer for recognizing our work. MFG4AD combines physics-based modeling and deep learning in a novel way to track tau spread directly on the brain surface. It not only predicts future tau accurately but also reveals interpretable Aβ–tau interactions, showing strong results on ADNI and OASIS datasets. Below, we address each of your concerns in detail.**
>
> **Q1: limited direct qualitative or quantitative evidence to substantiate how interpretable the learned laws**
>
> **A1:** Thank you for highlighting the need for more explicit evidence supporting the interpretability of the learned Aβ–tau interaction law. In response, we have revised Section 3.2 (line 298-320) to provide a detailed examination of the symbolic expressions $R_j(u,v)$ generated by our model.
>
> To demonstrate mechanistic interpretability, we present a representative symbolic reaction equation at a vertex in the medial parahippocampal cortex—a region known to initiate tau pathology (Braak I–II) [Braak & Braak, 1991]. The symbolic expression incorporates terms from spatially and functionally linked Aβ sources, including the anterior/posterior cingulate cortex and frontal–parahippocampal junction, consistent with known structural pathways and disease staging trajectories. Each term in the symbolic expression is biologically meaningful:
>
> - Positive coefficients (e.g., $+0.02*v_{81}$​) denote facilitatory effects from local or upstream Aβ burden. For instance, anterior cingulate cortex ($v_{81}$) has been identified as an early Aβ hub in Thal Phase 2 and is tightly linked to the medial temporal lobe via the cingulum bundle [Thal et al., 2002,Neurology; Jacobs et al., 2018,Neuron].
>
> - Nonlinear components (e.g., $0.2\sin(2.2v_9) + 1.9 + 1.9\exp(3.5v_{16}) + 1$) capture thresholded or saturating influences of Aβ, reflecting empirical findings that Aβ’s effect on tau becomes pronounced only beyond certain levels [Hanseeuw et al., 2019, JAMA Neurology]. These forms also simulate sigmoid-like molecular switch behavior, analogous to pathological tipping points in disease cascades.
>
> - Negative weights (e.g., $-0.09u_{44}$​) arising from regions such as the calcarine (visual cortex) suggest inhibitory or damping effects, consistent with their involvement in later Braak stages (V–VI) and possible negative feedback or non-contributory roles in early tau spread [Busche & Hyman, 2020,Nature Neuroscience].
>
> Importantly, the spatial configuration of contributing terms reflects a biologically coherent propagation route—the entorhinal–hippocampal–cingulate loop, widely regarded as the epicenter of early Alzheimer’s pathology [Braak & Braak, 1991; Jacobs et al., 2018].
>
> Together, these findings demonstrate that the learned symbolic laws are not only mathematically predictive, but also biologically interpretable, aligning with canonical disease progression stages and known anatomical connectivity.
>
> **Q2: insufficient technical details regarding its symbolic regression module**
>
> **A2** Thank you for this important point. We have provided more detailed implementation of the symbolic regression module. Specifically:
>
> - Architecture: The symbolic module (*SymbolicNetL00*) is composed of multiple *SymbolicLayerL0* layers (as detailed in the anonymous GitHub). Each layer transforms inputs using a collection of interpretable mathematical functions (e.g., identity, sin, exp, product) and learns to select sparse combinations through a differentiable binary mask.
>
> - Search space of functions: The model uses a library of symbolic primitives including: Unary functions: identity, $\sin(x)$, $\cos(x)$, $\exp(x)$, etc. Binary functions: product $x \cdot y$, quotient $x/y$, etc.
>
> - Optimization & regularization: We employ the hard concrete distribution as in [Louizos et al., 2017] to sample differentiable binary masks $z \in [0,1]$ applied element-wise to the weight matrix. This enables $L0$ sparsity regularization, encouraging simpler and more interpretable symbolic formulas.
>
> - Selection criteria: After training, symbolic terms with high dropout probabilities (i.e., low expected activation z) are pruned to remove weak or unstable components. The remaining active terms are ranked based on the magnitude of their learned weights, and their corresponding expressions are reconstructed using a predefined set of interpretable mathematical functions (e.g., linear, sine, exponential), ensuring biological and mathematical interpretability.
>
> - Robustness & Reproducibility: We evaluated the learned symbolic expressions using 5-fold cross-validation. Key contributing regions—such as the entorhinal cortex—consistently ranked among the top terms across folds. Fig. 6 summarizes the selection frequency of each Aβ region, highlighting the stability and biological plausibility of the learned interaction patterns. We will make it clear in the final version, thanks.
>
> **Q3: the sensitivity of the model's performance to variations in these hyperparameters**
>
> **A3:** Thank you for your comment, we have included the sensitivity analysis of hyperparameter choices (line 203-208 in the manuscript).
>
> | $η\_C\|η\_G$ | $n\_C$ | gp | λ | MAE       | RMSE       |
> | ---------- | ---- | -- | ---------- | --------------- | --------------- |
> | 1e-5\|1e-4 | 5    | 10 | 1          | 0.3911 ± 0.1521 | 0.5505 ± 0.2551 |
> | 1e-5\|1e-4 | 5    | 10 | 5          | 0.0696 ± 0.0052 | 0.0976 ± 0.0063 |
> | 1e-5\|1e-4 | 5    | 10 | 10         | **0.0643 ± 0.0044** | **0.0933 ± 0.0094** |
> | 1e-5\|1e-4 | 5    | 10 | 15         | 0.0664 ± 0.0049 | 0.0945 ± 0.0059 |
> | 1e-5\|1e-4 | 5    | 10 | 20         | 0.0671 ± 0.0045 | 0.0952 ± 0.0055 |
> | 1e-4\|1e-4       | 5    | 5  | 10         | 0.0667 ± 0.0043 | 0.0947 ± 0.0053 |
> | 1e-4\|1e-4       | 5    | 10 | 10         | 0.0675 ± 0.0052 | 0.0955 ± 0.0062 |
> | 1e-4\|1e-4       | 5    | 15 | 10         | 0.0674 ± 0.0052 | 0.0955 ± 0.0062 |
> | 1e-4\|1e-4      | 3    | 10 | 10         | 0.0667 ± 0.0044 | 0.0947 ± 0.0054 |
> | 1e-4\|1e-4       | 7    | 10 | 10         | 0.0675 ± 0.0051 | 0.0955 ± 0.0061 |
> | 5e-5\|5e-5       | 5    | 10 | 10         | 0.0674 ± 0.0052 | 0.0954 ± 0.0062 |
> | 5e-4 \|5e-4      | 5    | 10 | 10         | 0.0668 ± 0.0045 | 0.0947 ± 0.0056 |
> | 1e-3\|1e-3       | 5    | 10 | 10         | 0.0678 ± 0.0048 | 0.0957 ± 0.0058 |
>
> - A smaller λ fails to adequately enforce reconstruction fidelity, leading to suboptimal predictions. The best performance is observed when $λ \in [10, 15]$, indicating this range strikes a desirable balance between adversarial realism and numerical accuracy.
> - Gradient penalty (gp) shows mild sensitivity across 5–15, suggesting the 1-Lipschitz constraint is sufficiently enforced within this range without overly penalizing the critic’s learning.
> - Critic steps ($n_C$) shows no monotonic trend; moderate values (e.g., 5) suffice to balance critic convergence and training efficiency.
> - Learning rates ($\eta_C = 1\text{e-5}, \eta_G = 1\text{e-4}$) yield the best result, as they support stable adversarial updates. Higher rates destabilize optimization, while lower rates may slow convergence.
>
> Training time is 74.02s/it, inference time is 0.23 s, on H100 NVL with 8 GPUs (94 G/per).
>
> **Q4: the biological assumptions about tau propagation, and empirical evidence or analysis show how the derived reaction laws**
>
> **A4:** Thank you for the question. Our model treats each cortical vertex as an agent whose tau level evolves under local interactions, governed by a surface-based brain graph. This graph is constructed directly from the native cortical surface mesh, preserving geodesic continuity and the anatomical folding pattern of the cortex (see Appendix A.1, Fig. 7). Unlike SC/FC-based connectomes, our geometry-aware network ensures that tau propagation follows biologically realistic, anatomically constrained trajectories.
>
> The weak coupling assumption in MFG aligns well with the biological reality: tau spread in one region is influenced not by a single source but by the average tau level across its neighboring regions. This naturally captures the notion of local accumulation and distributed influence along the cortical mantle. We adapt the MFG framework by integrating spatial connectivity into the interaction terms, such that the propagation dynamics respect surface-based neighborhood structures.
>
> The learned reaction laws provide not only enhanced predictive capability but also interpretable symbolic expressions, which allows us to quantify how local Aβ burden and spatially adjacent influences drive tau accumulation. These symbolic laws highlight early-stage epicenters (e.g., entorhinal and medial prefrontal cortex) and map out downstream propagation patterns consistent with Braak I–III staging. By coupling biophysically grounded diffusion, geometric network constraints, and interpretable equations, our model offers novel, data-driven insights into the spatial logic of tau spread in Alzheimer's disease.
>
> **Q5: lacks a comprehensive discussion**
>
> **A5:** Thank you for this important question. Beyond predictive accuracy, our model generates interpretable diffusion fields that offer biological insights into tau propagation dynamics. In Fig. 5, we visualize the spatial propagation of tau across clinical stages and datasets (ADNI and OASIS). The inferred flow fields $\nu$ reveal coherent directional patterns that mirror canonical staging: for instance, early spread originates in the medial temporal lobe (e.g., entorhinal cortex), progressing toward isocortical regions such as parietal and frontal cortices—consistent with Braak stages I–VI.
>
> Importantly, these flow fields were derived purely from data via our model, without imposing priors about direction or staging. The emergence of biologically plausible propagation axes, repeatedly observed across datasets and stages, demonstrates that our framework captures not just the distribution but also the mechanism of tau spread. This provides novel empirical support for network-based propagation hypotheses in AD.

---

> > ### Comment · Area_Chair_ZXUK · 2025-08-05
> > **Author-Reviewer Discussion Reminder**
> >
> > Dear Reviewer mzBH,
> >
> > As the deadline for author-reviewer discussion is approaching, could you please check the authors' rebuttal and post your response?
> >
> > Thank you!
> >
> > Best,
> >
> > AC

---

> > ### Comment · Reviewer_mzBH · 2025-08-07
> >
> > I appreciate the authors’ detailed rebuttal and clarifications. I will maintain my current score.

---

> > > ### Author Response · Authors · 2025-08-07
> > >
> > > We sincerely appreciate your time and efforts of acknowledging our response.

---

> ### Author Response · Authors · 2025-08-07
> **Follow-up Request for Reviewer mzBH (Rebuttal Deadline Approaching)**
>
> Dear Reviewer mzBH,
>
> We would like to sincerely thank you for the time, effort, and expertise you devoted to reviewing our submission. Your thoughtful feedback, particularly regarding the hyperparameter setting analysis and interpretability of the learned laws, has helped us significantly improve the quality of our work. We are truly grateful for the opportunity to improve our work through this revision process.
>
> We hope that our rebuttal has adequately addressed your concerns. **If so, we would be most grateful if you would consider updating your score accordingly.**
>
> Of course, if you have any remaining concerns or further suggestions, please do not hesitate to let us know, we would be more than happy to address them to the best of our ability.
>
> With sincere appreciation,
>
> Authors

---

### Official Review · Reviewer_ACia · 2025-07-21

**Clarity:** 3
**Significance:** 4
**Originality:** 4
**Rating:** 4
**Confidence:** 4

**Summary:**

They propose a physics-informed deep learning framework that unites biophysical modeling and data-driven discovery to reconstruct tau propagation dynamics from longitudinal tau PET scans. This paper formulates the dynamics of tau propagation as a mean-field game (MFG). By leveraging the variational primal-dual structure in MFG, they propose a Wasserstein-1 Lagrangian generative adversarial network (GAN), in which a Lipschitz critic seeks the appropriate transport cost at the population level and a generator parameterizes the flow fields of optimal transport across individuals. Additionally, they incorporate a symbolic regression module to derive an explicit formulation capturing the Aβ-tau crosstalk.

**Questions:**

When you show the prediction performance in table 1, do you use all of the samples? How is the performance in EMCI, LMCI, and AD, respectively? What is the difference between CN, MCI, and AD? Can you also demonstrate the correlation between the truth and the predicted one? Can we get the same conclusion?
Is there any reference to support the new insights of Abeta Tau interaction in AD?
Is there any statistical result to demonstrate similar findings in tau propagation and interaction between ADNI and OASIS datasets?

**Ethical Concerns:**

["NO or VERY MINOR ethics concerns only"]

**Limitations:**

The paper does not provide performance metrics for specific subgroups such as EMCI, LMCI, and AD.
The study does not demonstrate the correlation between predicted and true values.. This may limit the interpretability and robustness of the findings.
The manuscript introduces new claims about Abeta-Tau interactions and tau propagation across datasets (ADNI and OASIS) without citing relevant literature or providing statistical evidence to support consistency between datasets.

**Quality:**

3

**Strengths And Weaknesses:**

This work models tau spread as a network-constrained reaction–diffusion process with a data-driven symbolic law for tau–amyloid crosstalk; casts this system as an equivalent potential mean field game, linking classical PDE theory to tau propagation; and employs a Wasserstein-1 Lagrangian GAN to learn optimal transport flows for accurate tau forecasting. On ADNI and OASIS cohorts, MFG4AD delivers state-of-the-art predictions for unseen subjects and resolves tau-flow directions, while also uncovering an explicit, interpretable reaction law, offering a powerful combination of predictive performance and mechanistic insight into Alzheimer’s pathology.

However, the paper does not provide performance metrics for specific subgroups such as EMCI, LMCI, and AD. The study does not demonstrate the correlation between predicted and true values.. This may limit the interpretability and robustness of the findings. The manuscript introduces new claims about Abeta-Tau interactions and tau propagation across datasets (ADNI and OASIS) without citing relevant literature or providing statistical evidence to support consistency between datasets.

---

> ### Author Rebuttal · Authors · 2025-07-30
>
> **Thank you for your thoughtful summary and evaluation of our work. We greatly appreciate your recognition of our framework’s strengths—including its principled integration of biophysical modeling and deep learning, the novel MFG-based formulation, and its potential to yield both accurate predictions and interpretable mechanistic insights. Below, we address each of your concerns in detail.**
>
> **Q1: Subgroups' performance**
>
> **A1**: We appreciate the reviewer’s attention to subgroup-level performance. In the original manuscript, we provided visual comparisons of model performance and tau propagation across diagnostic subgroups (CN, EMCI, LMCI, and AD), as shown in Fig. 5  (main text) and Fig. 8 (Appendix). Table 1 summarizes the performance on all samples. Thank you for reminding us and this is a very helpful suggestion. We have now included the quantitative performance metrics (e.g., MAE, RMSE) for each subgroup. These results complement the original table and provide a more rigorous and transparent evaluation of model performance across disease stages. We will incorporate all the results in the final version.
>
>
> |  **ADNI**    | CN\|SMC            | EMCI                | LMCI                | AD                  |
> | ---- | ------------------ | ------------------- | ------------------- | ------------------- |
> | MAE  | 0.0727 ± 0.033 | 0.0600 ± 0.0253 | 0.0628 ± 0.0086 | 0.0617 ± 0.0295 |
> | RMSE | 0.1282 ± 0.029 | 0.0757 ± 0.0207 | 0.0787 ± 0.0072 | 0.0790 ± 0.0147 |
>
> We also note that the CN group in our analysis includes subjects with Subjective Memory Complaint (SMC) in ADNI dataset. While clinically categorized as cognitively normal, SMC individuals often present subtle biomarker changes or early cognitive concerns. Their inclusion likely increases the heterogeneity within the CN group and may contribute to the slightly higher prediction error observed in this subgroup. We have clarified this point in the revised manuscript.
>
> |  **OASIS**     | CN                 | AD                 |
> | ---- | ------------------ | ------------------ |
> | MAE  | 0.4289 ± 0.065 | 0.4919 ± 0.075 |
> | RMSE | 0.6104 ± 0.050  | 0.7041 ± 0.130  |
>
> **Q2: Relevant literature about Abeta-Tau interactions and, quantitative cross‑dataset evidence**
>
> **A2**: We appreciate this observation and have revised the manuscript to provide both (i) supporting literature and (ii) quantitative cross‑dataset evidence.
> - We now cite several large‑scale imaging and modelling studies demonstrating that local Aβ burden amplifies connectivity‑constrained tau propagation: [1] Lee et al. “Regional A-tau interactions promote onset and accelertion of Alzheimer’s disease tau spreading”, Neuron, 2022. [2] Roemer-Casiano et al. “Amyloid-associated hyperconnectivity drives tau spread across connected brain regions in Alzheimer’s disease”, Science Translatioal Medicine, 2025
> - To demonstrate cross‑dataset consistency, we computed vertex‑wise Aβ–tau interaction frequency maps for ADNI (64,262 left‑ and 54,128 right‑hemisphere vertices) and OASIS (34,917 LH / 26,898 RH) and then summed these frequencies within the 160 Destrieux regions of interest [Destrieux, C., et al. Automatic parcellation of human cortical gyri and sulci using standard anatomical nomenclature. Neuroimage]. This procedure identified 67 high‑frequency ROIs in ADNI and 37 in OASIS; 22 regions (24663 vertices)—bilateral entorhinal, parahippocampal, fusiform, inferior/middle temporal, posterior cingulate/precuneus, caudate, and amygdala—were common in both cohorts, encompassing more than 20 000 vertices (Dice = 0.42). Crucially, the Aβ‑related frequencies in these 22 overlapping parcels remained strongly concordant across datasets (Spearman ρ = 0.46). A 10,000‑iteration permutation test that randomly reassigned OASIS ROI labels produced an empirical p‑value of 0.009, confirming that such concordance is exceedingly unlikely to occur by chance and reinforcing the cross‑dataset reproducibility of the spatial pattern of Aβ–tau synergy observed in ADNI and OASIS.
>
> We will incorporate all the results in the final version.
>
> **Q3:  The correlation between the truth and the predicted one**
>
> **A3:** Thank you for your comment. We have added a dedicated figure (Fig. R1) that visualises predicted vs. observed tau‑SUVR means for every diagnostic subgroup in each cohort. Each dot is an individual subject, the numerical results are summarised here:
> ### Predicted vs. Observed Tau-SUVR Fitting Metrics
>
> | Cohort / Subgroup | Slope         | R²             |
> |-------------------|---------------|----------------|
> | **ADNI (overall)**| 0.98 ± 0.06   | 0.91 ± 0.07    |
> | └── CN            | 1.055         | 0.822          |
> | └── EMCI          | 0.975         | 0.953          |
> | └── LMCI          | 1.001         | 0.895          |
> | └── AD            | 1.125         | 0.982          |
> | **OASIS (overall)**| 0.90 ± 0.16  | 0.660 ± 0.17   |
> | └── CN            | 1.007         | 0.785          |
> | └── AD            | 0.784         | 0.535          |
>
> These metrics demonstrate an agreement between predicted and true values across all subgroups. We have prepared the corresponding fitting curves (Rebuttal_Fig.R1.pdf) to support verification. However, due to the rebuttal policy restricting supplementary uploads during the review process, we are unable to provide this figure at this stage. Nevertheless, we will incorporate these results into the final version of the manuscript, as we agree that they provide important support for the paper. We sincerely appreciate your suggestion.
>
> **Thank you for your time and consideration.**

---

> > ### Comment · Area_Chair_ZXUK · 2025-08-05
> > **Author-Reviewer Discussion Reminder**
> >
> > Dear Reviewer ACia,
> >
> > As the deadline for author-reviewer discussion is approaching, could you please check the authors' rebuttal and post your response?
> >
> > Thank you!
> >
> > Best,
> >
> > AC

---

> > > ### Author Response · Authors · 2025-08-07
> > > **Follow-up Request for Reviewer ACia (Rebuttal Deadline Approaching)**
> > >
> > > Dear Reviewer ACia,
> > >
> > > We would like to sincerely thank you for the time, effort, and expertise you devoted to reviewing our submission. Your thoughtful feedback, particularly regarding the subgroups' performance analysis, has helped us significantly improve the quality of our work. We are truly grateful for the opportunity to improve our work through this revision process.
> > >
> > > We hope that our rebuttal has adequately addressed your concerns. **If so, we would be most grateful if you would consider updating your score accordingly.**
> > >
> > > Of course, if you have any remaining concerns or further suggestions, please do not hesitate to let us know, we would be more than happy to address them to the best of our ability.
> > >
> > > With sincere appreciation,
> > >
> > > Authors

---

### Note · Authors · 2025-08-11

Dear ACs, SACs and Reviewers,

We would like to thank all reviewers, area chairs, and organizers for their tireless efforts. We are glad that reviewers have shown interest in our *MFG4AD*, as evidenced by the positive ratings and the remarks from reviewers.

The comments and suggestions provided by the reviewers are remarkably constructive, straightforward, and easily fixed, enabling us to make substantial enhancements to the paper's overall quality. We are committed to incorporating all discussions and updates in the final version.

Our work integrates **mean-field games, optimal transport, GANs, and symbolic regression into a unified physics-informed framework, presenting a novel and innovative approach to modeling Alzheimer’s disease pathology. This approach bridges biophysical modeling with data-driven discovery effectively** (recognized by all reviewers). We firmly believe that this contribution is of significant importance and merits dissemination at the prestigious NeurIPS.

Thank you for your attention.

Best,

Authors

---

### Decision · Program_Chairs · 2025-09-17

**Decision:**

Accept (poster)

**Comment:**

This paper a novel physics-informed deep learning framework to model tau propagation in Alzheimer's disease. The proposed method adopts a graph-based cortical mesh, a GAN to learn transport flows, and a symbolic regression module to learn how amyloid affects tau. Experiments on public neuroimaging datasets show that the proposed method not only provides precise and reliable predictions of future tau accumulation but also reveals interpretable dynamics consistent with recent imaging studies.

All of the reviewers recognized the novelty and technical contributions of this work. The overall formulation is novel, e.g., combining MFGs and Wasserstein GANs. Also, the idea of incorporating symbolic regression to learn explicit, interpretable tau - Aβ reaction laws is quite interesting. In addition, experiments are comprehensive and convincing, and results show that the proposed method outperforms several strong baselines.

Meanwhile, reviewers raised some questions regarding assumptions, technical details, paper presentation, experiments, etc. The authors have provided detailed responses with additional results, which have addressed the previous concerns from reviewers. The authors are strongly encouraged to incorporate the new results and discussions into the final version of the paper.